# Learning-Augmented Facility Location Mechanisms for Envy Ratio

**Haris Aziz**
UNSW Sydney
haris.aziz@unsw.edu.au

**Yuhang Guo**
UNSW Sydney
yuhang.guo2@unsw.edu.au

**Alexander Lam**
Hong Kong Polytechnic University
alexander-a.lam@polyu.edu.hk

**Houyu Zhou**[*]
UNSW Sydney
houyu.zhou@unsw.edu.au

## Abstract

The augmentation of algorithms with predictions of the optimal solution, such as from a machine-learning algorithm, has garnered significant attention in recent years, particularly in facility location problems. Moving beyond the traditional focus on utilitarian and egalitarian objectives, we design learning-augmented facility location mechanisms on a line for the envy ratio objective, a fairness metric defined as the maximum ratio between the utilities of any two agents. For the deterministic setting, we propose a mechanism which utilizes predictions to achieve $\alpha$-consistency and $\frac{\alpha}{\alpha-1}$-robustness for a selected parameter $\alpha \in [1, 2]$, and prove its optimality. We also resolve open questions raised by Ding et al. [2020], devising a randomized mechanism without predictions to improve upon the best-known approximation ratio from 2 to $1.8944$. Building upon these advancements, we construct a novel randomized mechanism which incorporates predictions to achieve improved performance guarantees.

## 1 Introduction

In the uni-dimensional facility location problem, we are given a set of agents who are located along an interval, and are tasked with finding an ideal location to place a facility. Each agent has single-peaked preferences, preferring the facility to be located as close to them as possible. Due to its simplicity as a continuous, single-peaked preference aggregation problem, the facility location problem has many industrial and societal applications, such as school/library/hospital placement [Schummer and Vohra, 2002], social/economic policy selection [Dragu and Laver, 2019] and budget aggregation [Freeman et al., 2021]. Among these applications, it is particularly important to achieve a *fair* solution, in which no agent (or subset of agents) is excessively distant from the facility. As a result, this problem has been widely studied in operations research, microeconomics, and theoretical computer science, with numerous papers proposing various solution concepts.

In social choice problems, a common egalitarian fairness concept is to maximize the utility/well-being of the *worst-off* agent, and when translated to the facility location problem, this is equivalent to the minimizing the worst-off agent's distance from the facility. This is achieved by placing the facility at the midpoint of the left- and right-most agents, but as Mulligan [1991] remarks, this solution is prone to perturbations of the extreme agent locations. Accordingly, numerous papers (e.g. [McAllister, 1976, Marsh and Schilling, 1994]) have proposed alternative fairness objectives which provide an improved measure of the inequality within an instance.

---

[*]Corresponding Author.

Our paper's focus is on the *envy ratio* objective, which was proposed for the facility location problem by Ding et al. [2020] and later expanded upon by Liu et al. [2020]. Informally speaking, the envy ratio of an instance is defined as the largest ratio between any two agents' utilities, in which each agent's utility is equal to the difference between the length of the domain and their distance from the facility. Unlike the maximum distance objective, the envy ratio objective is a pairwise fairness notion which additionally takes into account the relative well-being of the best-off agent. To further illustrate the difference and added nuance, consider an instance with two agents, located at the extremities of the interval. When the facility is placed at the midpoint, the addition of new agents near the midpoint does not affect the maximum distance, but causes the envy ratio to increase, as the agents at the endpoints become envious of the agents who are located near the facility. When considering instances that correspond to a large approximation ratio, the envy ratio's dependence on the agents' utilities creates a comparatively larger focus on instances where agents' receive low welfare from the mechanism, such as when they are located at the domain endpoints. Importantly, the envy ratio objective also respects fundamental fairness principles such as the Pigou-Dalton principle [Sen, 1997] and the Rawlsian Principle [Rawls, 2017].

Aside from fairness concerns, we typically desire a solution which is additionally *strategyproof*, incentivizing agents to reveal their true locations by ensuring that any misreporting is not beneficial. This is important when the agents' locations are assumed to be *private* information, and is the defining goal in the extensive literature on approximate mechanism design [Moulin, 1980, Procaccia and Tennenholtz, 2013]. Essentially, the aim is to design strategyproof mechanisms which have a bounded approximation for some (typically utilitarian or egalitarian) objective, and to compute a lower bound on the best-possible approximation by a strategyproof mechanism. For the envy ratio objective, Ding et al. [2020] show that deterministic strategyproof mechanisms can achieve a 2-approximation at best, and give bounds on the approximation achievable by randomized mechanisms.

In this work, we ask whether we can improve upon the best-known envy ratio approximation results by designing mechanisms which additionally utilize a *prediction* of the optimal facility placement (such as from a machine-learning algorithm trained on historical data). This approach stems from the field of *learning-augmented* algorithms, which has seen significant interest in recent years, as the additional prediction information is leveraged to improve upon the traditional 'worst-case' approximation ratio bounds. In this context, an ideal mechanism provides an outcome which is close to the social optimum when the predictor provides accurate information (*consistency*), and also retains worst-case approximation guarantees when the predictor is inaccurate (*robustness*).

## 1.1 Our Results

In this paper, we apply learning augmentation to design anonymous and strategyproof mechanisms which have a bounded approximation ratio for the envy ratio objective. Our main results are as follows.

1. In the deterministic setting, we propose the novel $\alpha$-Bounding Interval Mechanism ($\alpha$-BIM), where $\alpha \in [1, 2]$ serves as a tunable parameter based on the confidence level of the prediction. This mechanism obtains $\alpha$-consistency and $\frac{\alpha}{\alpha-1}$-robustness.

2. We demonstrate the optimality of the $\alpha$-BIM mechanism by showing that no deterministic, strategyproof, and anonymous mechanism can achieve $(\alpha - \varepsilon)$-consistency and $(\frac{\alpha}{\alpha-1} - \varepsilon)$-robustness. We further explore fine-grained approximation ratios parameterized by the prediction error, which smoothly transition from $\alpha$-consistency to $\frac{\alpha}{\alpha-1}$-robustness as the error bound increases.

3. We next revisit open problems relating to randomized mechanisms *without* predictions. We first introduce a class of randomized mechanisms termed $(\alpha, p)$-LRM constant mechanisms, and prove that the $(\frac{\sqrt{5}}{2} - 1, \frac{2}{5})$-LRM constant mechanism achieves a $1 + \frac{2}{\sqrt{5}} \approx 1.8944$-approximation w.r.t. envy ratio, which is optimal within the family of $(\alpha, p)$-LRM mechanisms. Our results resolve an open question posed by Ding et al. [2020], which asked whether a randomized mechanism with an approximation ratio strictly below 2 could be found.

4. We show that any randomized mechanism without predictions has an approximation ratio of at least 1.1125, improving the best-known lower bound from 1.0314.

5. To address the challenging problem of devising randomized mechanisms with predictions, we propose the Bias-Aware Mechanism (BAM), in which the probability distribution of the facility

depends on the deviation of location prediction $\hat{y}$ from the interval midpoint $\frac{1}{2}$. We demonstrate that BAM achieves $(-4c^2 + 2)$-consistency and $(c + 2)$-robustness when $c \in [\frac{1}{4}, \frac{1}{2}]$, and $\frac{7}{4}$-consistency and $\frac{9}{4}$-robustness when $c \in [0, \frac{1}{4})$, where $c = |\hat{y} - \frac{1}{2}|$. BAM strictly outperforms the deterministic $\alpha$-BIM in terms of both consistency and robustness performance guarantees. In addition to BAM, we investigate other potential mechanisms to provide a broader perspective on the learning-augmented mechanism design. Results lacking full proofs are proven in the appendix.

## 1.2 Related Work

**Fairness in Facility Location Mechanism Design**   Fairness concerns and objectives have been long-studied in facility location problems; early works in operations research (e.g., [McAllister, 1976, Marsh and Schilling, 1994, Mulligan, 1991]) discuss optimal solutions for fairness objectives such as the Gini coefficient and the mean deviation, quantifying various inequity notions. On the other hand, the seminal paper by Procaccia and Tennenholtz [2009] introduces an approximate mechanism design approach, in which they design strategyproof facility location mechanisms with a bounded approximation ratio for various objectives, including the cost/distance incurred by the worst-off agent. This measure of egalitarian fairness has a similar underlying principle as our *envy ratio* objective, which represents, in a multiplicative sense, the envy that the worst-off agent has for the best-off agent. This objective was introduced for the one-facility location problem by Ding et al. [2020], and later extended to multiple facilities in the subsequent work [Liu et al., 2020]. A similar notion of minimax envy quantifies the envy in an additive sense (i.e., the maximum difference between any two agents' distances from the facility), and was studied by Cai et al. [2016] and Chen et al. [2022]. Walsh [2025] studied Gini index in facility location mechanism design.

Other than egalitarian/worst-off fairness, fairness objectives relating to groups of agents can be considered. For instance, Zhou et al. [2022, 2024] consider two group-fair objectives, the maximum total cost incurred by a group of agents, and the maximum average cost incurred by a group of agents. One may also consider representing group fairness via axioms that must be satisfied by a 'group-fair' mechanism. For instance, distance/utility guarantees can be imposed for endogenously defined groups of agents at or near the same location, in which the magnitude of the guarantee is proportional to the size of the group [Aziz et al., 2022, Lam et al., 2024, Aziz et al., 2025]. For other related work and variations of facility location problems, we refer the reader to a recent survey by Chan et al. [2021].

**Facility Location Mechanisms with Predictions**   Research on facility location mechanisms which are augmented with predictions has flourished in recent years, beginning with the paper by Agrawal et al. [2022], which studies deterministic mechanisms that take a prediction of the optimal facility location as an additional input. They provide *best-of-both-worlds* style results, designing mechanisms which perform (in terms of their approximation ratio) *consistently* well when an accurate prediction is provided, and are *robust* to entirely inaccurate predictions. The concept of measuring both the consistency and robustness of learning-augmented algorithms was first introduced by Lykouris and Vassilvitskii [2021], and it has since been applied in numerous extensions of facility location problems. Balkanski et al. [2024] studied randomized mechanisms for the egalitarian/maximum cost objective, whilst Chen et al. [2024] extend the domain to a continuous general metric space. For the two-facility location problem, Barak et al. [2024] design randomized mechanisms which use *mostly* and *approximately* correct (MAC) predictions of the agents' locations, supplementing the work by Xu and Lu [2022], which explores a deterministic mechanism for the same problem. Aside from facility location problems, learning-augmented algorithms have been used for a wide variety of settings. For additional references, readers may refer to the ALPS website [Lindermayr and Megow, 2022].

## 2 Preliminaries

For any $t \in \mathbb{N}$, denote $[t] := \{1, 2, \ldots, t\}$. Let $N = [n]$ be a set of agents, where each agent $i$ has a location[2] $x_i \in [0, 1]$. We denote the location profile of $N$ by $\mathbf{x} = (x_1, x_2, \ldots, x_n) \in [0, 1]^n$. A *deterministic mechanism* $f : [0, 1]^n \to [0, 1]$ takes a location profile $\mathbf{x}$ as input, and outputs a facility location $y \in [0, 1]$, under which each agent $i \in N$ has a utility of $u(y, x_i) = 1 - d(y, x_i)$, where $d(y, x_i) = |y - x_i|$ represents the distance between the facility $y$ and $i$'s location. A *randomized*

---

[2]Our results hold w.l.o.g. for any compact interval domain.

*mechanism* $f : [0,1]^n \to \Delta([0,1])$ maps the location profile $\mathbf{x}$ to a probability distribution $P$ over $[0,1]$, under which each agent $i \in N$ has an expected utility of $u(P, x_i) = 1 - \mathbb{E}_{y \in P}[|y - x_i|]$.

We are primarily focused with the *envy ratio* objective, formally defined for an instance as the ratio between the best-off and worst-off agents' utilities.[3]

**Definition 2.1** (Envy Ratio). Given a location profile $\mathbf{x} \in \mathbb{R}^n$ and mechanism $f$, the envy ratio is

$$\mathrm{ER}(f(\mathbf{x}), \mathbf{x}) = \max_{i \neq j} \frac{u(f(\mathbf{x}), x_i)}{u(f(\mathbf{x}), x_j)}.$$

For a given mechanism $f$, we can quantify its worst-case performance (over all possible location profiles) with respect to the envy ratio objective via its *approximation ratio*.

**Definition 2.2** (Approximation Ratio). A mechanism $f$ is said to have an approximation ratio of $\rho$ if

$$\rho = \sup_{\mathbf{x} \in [0,1]^n} \frac{\mathrm{ER}(f(\mathbf{x}), \mathbf{x})}{\mathrm{ER}(\mathrm{OPT}(\mathbf{x}), \mathbf{x})},$$

where $\mathrm{OPT}(\mathbf{x})$ is the optimal solution which minimizes the envy ratio for any given location profile $\mathbf{x} \in [0,1]^n$. For any specific instance $\mathbf{x}$, let $\rho(\mathbf{x})$ denote the approximation ratio of $f$ under $\mathbf{x}$.

Note that $\rho \geq 1$ for all $f$. As proven by Ding et al. [2020], the OPT mechanism is the well-known midpoint mechanism which places the facility halfway between the left-most and right-most agent locations.

**Lemma 2.3** (Ding et al. [2020]). *Given any location profile instance $\mathbf{x}$, the midpoint mechanism $f(\mathbf{x}) = \mathrm{mid}(\mathbf{x}) = \frac{\mathrm{lm}(\mathbf{x}) + \mathrm{rm}(\mathbf{x})}{2}$ (where $\mathrm{lm}(\mathbf{x}) := \min_{i \in N}\{x_i\}$, and $\mathrm{rm}(\mathbf{x}) := \max_{i \in N}\{x_i\}$) optimizes the envy ratio objective.*

Throughout the paper, our proofs focus on the case where $\mathrm{lm}(\mathbf{x}) < \mathrm{rm}(\mathbf{x})$ and omit the case where $\mathrm{lm}(\mathbf{x}) = \mathrm{rm}(\mathbf{x})$, in which any feasible facility location $f(\mathbf{x})$ achieves an optimal envy ratio of 1.

As standard in facility location mechanism design, we assume that the agents' true locations are private information, and that the mechanism takes as input the locations which are *reported* by the agents. Accordingly, we restrict our attention to *strategyproof* mechanisms, which disincentivize agents from misreporting their location.

**Definition 2.4** (Strategyproofness). A mechanism $f$ is *strategyproof* if for any location profile $\mathbf{x} \in [0,1]^n$, we have $u(f(\mathbf{x}), x_i) \geq u(f(\mathbf{x}_{-i}, x_i'), x_i)$ for all $i \in N$ and $x_i'$.

Note that by definition, strategyproofness is defined in expectation if $f$ is a randomized mechanism.

In this paper, we discuss *learning-augmented* mechanisms, which take a prediction $\hat{y}$ of the optimal facility location as an additional input. We denote these mechanisms by $f(\mathbf{x}, \hat{y})$. Our goal is to design strategyproof mechanisms which have best-of-both-worlds approximation ratio guarantees, performing *consistently* well when $\hat{y}$ is a perfectly accurate prediction, and also being *robust* to inaccurate predictions of the optimal solution. Formally, we define the two performance metrics as follows.

**Definition 2.5** ($\gamma$-consistency). A mechanism $f$ is $\gamma$-*consistent* if it achieves an approximation ratio of $\gamma$ when given a *correct* prediction $\hat{y} = \mathrm{OPT}(\mathbf{x})$, i.e.,

$$\gamma = \sup_{\mathbf{x} \in [0,1]^n} \frac{\mathrm{ER}(f(\mathbf{x}, \mathrm{OPT}(\mathbf{x})), \mathbf{x})}{\mathrm{ER}(\mathrm{OPT}(\mathbf{x}), \mathbf{x})}.$$

**Definition 2.6** ($\beta$-robustness). A mechanism $f$ is $\beta$-*robust* if it achieves an approximation ratio of $\beta$ under *any* prediction $\hat{y}$, i.e.,

$$\beta = \sup_{\mathbf{x} \in [0,1]^n, \hat{y} \in [0,1]} \frac{\mathrm{ER}(f(\mathbf{x}, \hat{y}), \mathbf{x})}{\mathrm{ER}(\mathrm{OPT}(\mathbf{x}), \mathbf{x})}.$$

Note that the mechanism which always places the facility at the predicted location $\hat{y}$ is 1-consistent but has unbounded robustness. We also remark that if a mechanism does not admit a prediction as input and is $\rho$-approximate, then it is $\rho$-consistent and $\rho$-robust in the learning-augmented setting.

---

[3]Note that a utility-based formulation is necessary to define this objective, as a distance-based definition results in an unbounded envy ratio whenever the facility coincides with an agent's location.

# 3 Deterministic Mechanisms

We begin with the deterministic setting, in which any strategyproof mechanism without predictions is known to have an approximation ratio of at least 2 (Theorem 1 in [Ding et al., 2020]), and that this lower bound is matched by the constant-$\frac{1}{2}$ mechanism which always places the facility at $\frac{1}{2}$. By admitting a facility location prediction as an additional input, we are able to extend the Constant-$\frac{1}{2}$ mechanism to the following $\alpha$-Bounding Interval Mechanism, which defines an interval based on a parameter $\alpha \in [1, 2]$, and places the facility at the prediction $\hat{y}$ if it lies within this interval. Otherwise, the facility is placed at a boundary point of this interval.

---

**Mechanism 1** $\alpha$-Bounding Interval Mechanism ($\alpha$-BIM)

---

**Input:** Location profile $\mathbf{x}$, facility location prediction $\hat{y}$, and parameter $\alpha \in [1, 2]$.
**Output:** Facility location $f(\mathbf{x}, \hat{y})$.
  1: **if** $\hat{y} \in [1 - \frac{1}{\alpha}, \frac{1}{\alpha}]$ **then**
  2:     Return $f(\mathbf{x}, \hat{y}) \leftarrow \hat{y}$;
  3: **else if** $\hat{y} \in (\frac{1}{\alpha}, 1]$ **then**
  4:     Return $f(\mathbf{x}, \hat{y}) \leftarrow \frac{1}{\alpha}$;
  5: **else**
  6:     Return $f(\mathbf{x}, \hat{y}) \leftarrow 1 - \frac{1}{\alpha}$;
  7: **end if**

---

Note that the output of this mechanism ranges from $f(\mathbf{x}, \hat{y}) = \hat{y}$ when $\alpha = 1$, to $f(\mathbf{x}, \hat{y}) = \frac{1}{2}$ when $\alpha = 2$, and thus its performance ranges from 1-consistency and unbounded robustness to 2-consistency and 2-robustness. As we will show, the $\alpha$-Bounding Interval Mechanism specifically has $\alpha$-consistency and $\frac{\alpha}{\alpha-1}$-robustness. While we do not achieve a strict improvement over the 2-consistency and 2-robustness of the Constant-$\frac{1}{2}$ mechanism, the added flexibility from the $\alpha$ parameter enables the central decision maker to choose their desired consistency-robustness tradeoff depending on their confidence in the prediction accuracy. We also remark that the mechanism is additionally *anonymous*, meaning that the output is invariant under any permutation of the agents' labelings. Before analyzing the consistency and robustness of $\alpha$-BIM, we first introduce a crucial lemma which simplifies the space of location profiles which need to be considered.

**Lemma 3.1.** *For any instance $\mathbf{x}$, and a distribution of facility locations $P$, there always exists a 2-agent instance $\mathbf{x}' = (\mathrm{lm}(\mathbf{x}), \mathrm{rm}(\mathbf{x}))$ such that $\mathbb{E}_{y \in P} \left[ \frac{\mathrm{ER}(y, \mathbf{x})}{\mathrm{ER}(\mathrm{mid}(\mathbf{x}), \mathbf{x})} \right] \leq \mathbb{E}_{y \in P} \left[ \frac{\mathrm{ER}(y, \mathbf{x}')}{\mathrm{ER}(\mathrm{mid}(\mathbf{x}'), \mathbf{x}')} \right]$.*

The proof idea is that given any instance $\mathbf{x}$ and for any location $y \in P$, either $y \in [\mathrm{lm}(\mathbf{x}), \mathrm{rm}(\mathbf{x})]$ or $y \notin [\mathrm{lm}(\mathbf{x}), \mathrm{rm}(\mathbf{x})]$, we show that the approximation ratio under $\mathbf{x}$ is always upper-bounded by that under $\mathbf{x}' = (\mathrm{lm}(\mathbf{x}), \mathrm{rm}(\mathbf{x}))$. The complete proof is relegated to Appendix A.1.

By Lemma 3.1, when analyzing the performance of mechanisms, we only need to focus on 2-agent instances. We now formally prove the consistency and robustness of $\alpha$-BIM.

**Theorem 3.2.** *$\alpha$-BIM is anonymous, strategyproof, and satisfies $\alpha$-consistency and $\frac{\alpha}{\alpha-1}$-robustness.*

*Proof.* $\alpha$-BIM is trivially strategyproof and anonymous, as the output is independent of the agents' locations. We next move to the analysis of consistency and robustness. From Lemma 3.1 we only need to consider instances $\mathbf{x} = (x_1, x_2)$ with two agents where $x_1 < x_2$.

**(Consistency).** Consider an arbitrary 2-agent instance $\mathbf{x}$ in which $\hat{y}$ is accurate, i.e., $\hat{y} = \mathrm{mid}(\mathbf{x})$. If $\mathrm{mid}(\mathbf{x}) = \hat{y} \in [1 - \frac{1}{\alpha}, \frac{1}{\alpha}]$, $\alpha$-BIM trivially satisfies 1-consistency. There are two remaining cases: either $\mathrm{mid}(\mathbf{x}) = \hat{y} \in [0, 1 - \frac{1}{\alpha})$ or $\mathrm{mid}(\mathbf{x}) \in (\frac{1}{\alpha}, 1]$. Due to symmetry, it suffices to focus on the former case, in which $\alpha$-BIM returns $f(\mathbf{x}, \hat{y}) = 1 - \frac{1}{\alpha}$. Since $\hat{y} < 1 - \frac{1}{\alpha}$, the maximum utility achieved by the facility location $1 - \frac{1}{\alpha}$ is contributed by $x_2$, and the minimum utility achieved by the facility location $1 - \frac{1}{\alpha}$ is contributed by $x_1$. Moreover, the utility of the agent at $x_2$ is at most 1. The utility of the agent at $x_1$ is at least $1 - (1 - \frac{1}{\alpha}) = \frac{1}{\alpha}$. Hence, the consistency is at most

$$\gamma = \sup_{\mathbf{x} \in [0,1]^n} \frac{\mathrm{ER}(f(\mathbf{x}, \mathrm{mid}(\mathbf{x})), \mathbf{x})}{\mathrm{ER}(\mathrm{mid}(\mathbf{x}), \mathbf{x})} \leq \frac{1}{1 - (1 - \frac{1}{\alpha})} = \alpha.$$

For a matching lower bound, consider an instance with 2 agents located at $x_1 = 0$ and $x_2 = 1 - \frac{1}{\alpha}$. Here, Mechanism 1 places the facility at $1 - \frac{1}{\alpha}$, leading to a consistency of at least $\alpha$. Therefore, we conclude that $\alpha$-BIM achieves $\alpha$-consistency.

**(Robustness)**. Consider an arbitrary 2-agent instance $\mathbf{x}$, suppose the mechanism outputs $y$. Since $y \in [1 - \frac{1}{\alpha}, \frac{1}{\alpha}]$, the minimum utility is at least $1 - \frac{1}{\alpha}$ and the maximum utility is at most 1. Hence, the robustness is at most

$$\beta = \sup_{\mathbf{x} \in [0,1]^n, \hat{y} \in [0,1]} \frac{\mathrm{ER}(f(\mathbf{x}, \hat{y}), \mathbf{x})}{\mathrm{ER}(\mathrm{mid}(\mathbf{x}), \mathbf{x})} \leq \frac{1}{1 - \frac{1}{\alpha}} = \frac{\alpha}{\alpha - 1}.$$

For a corresponding lower bound, consider a 2-agent instance, with the agents located at $1 - \frac{1}{\alpha}$ and 1. The optimal facility location in this case would be $1 - \frac{1}{2\alpha}$, achieving an envy ratio of 1. If $\hat{y} \in [0, 1 - \frac{1}{\alpha})$, then the mechanism selects $1 - \frac{1}{\alpha}$ as the facility location, leading to an envy ratio (and therefore robustness lower bound) of $\frac{\alpha}{\alpha - 1}$. □

We next show that for the envy ratio objective, $\alpha$-BIM obtains the best possible consistency and robustness guarantees among all strategyproof and anonymous deterministic mechanisms, establishing the optimality of the mechanism.[4]

**Theorem 3.3** (Optimality). *Given any parameter $\alpha \in (1, 2]$, there is no deterministic, strategyproof, and anonymous mechanism that is $(\alpha - \varepsilon)$-consistent and $(\frac{\alpha}{\alpha - 1} - \varepsilon)$-robust with respect to envy ratio, for any $\varepsilon > 0$.*

*Proof.* By the characterization of Moulin [1980] (Proposition 2), a deterministic strategyproof and anonymous mechanism must be a phantom mechanism with $n + 1$ 'constant' points/phantoms. A phantom mechanism places the facility at the median of the $n$ agent points and the $n + 1$ constant points. Note that the 'phantom' locations may be a function of the prediction.

Observe that when $\alpha \in (1, 2]$, we have $1 - \frac{1}{\alpha} \leq \frac{1}{\alpha}$. Next, given any prediction $\hat{y}$, we will show that all $n + 1$ phantoms must be in $[1 - \frac{1}{\alpha}, \frac{1}{\alpha}]$ in order for the robustness to be $\frac{\alpha}{\alpha - 1}$ or better. To see this, suppose for contradiction that one of those phantoms (denoted by $p_i = f_i(\hat{y})$) is in $[0, 1 - \frac{1}{\alpha})$. Since $p_i$ only depends on $\hat{y}$, we have that $p_i \in [0, 1 - \frac{1}{\alpha})$ is a fixed point for every set of locations $x_i, \ldots, x_n$. Now consider a location profile with $n - 1$ agents at $p_i$ and one agent at 1. Under this location profile, the facility will be placed at $p_i$, which leads to an envy ratio of $p_i$ and implies an approximation ratio of at least $\frac{1}{p_i}$. Since $p_i < 1 - \frac{1}{\alpha}$, the robustness will be strictly greater than $\frac{\alpha}{\alpha - 1}$.

Next, for the same fixed $\hat{y}$, we consider another location profile with $n - 1$ agents at $\frac{1}{\alpha}$ and one agent at 1. The facility will be placed in the interval $[1 - \frac{1}{\alpha}, \frac{1}{\alpha}]$, leading to $\alpha$-consistency at best. Therefore, if the robustness is $\frac{\alpha}{\alpha - 1}$ or better, the consistency cannot be better than $\alpha$, proving the result. □

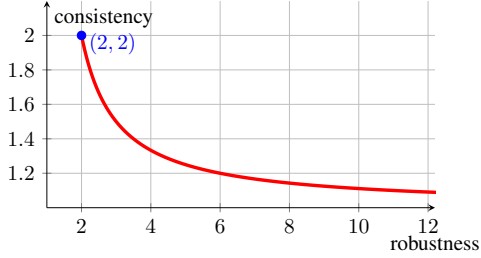

Figure 1: Trade-off between consistency and robustness under $\alpha$-BIM

We also depict the trade-off between consistency and robustness, as determined by the parameter $\alpha$, in Figure 1 below. One may adjust the parameter $\alpha$ according to the confidence of the prediction, i.e., setting a small (resp. large) $\alpha$ when the confidence in the prediction is high (resp. low).

---

[4]For $\alpha = 1$, it is trivial that no mechanism with 1-consistency can achieve bounded robustness.

**Approximation Ratio Parameterized by Prediction Error**   We now extend the consistency and robustness results for $\alpha$-BIM to obtain a refined approximation ratio parameterized by the prediction error. Let $y^*$ denote the optimal facility location $\text{OPT}(\mathbf{x})$ and $\eta$ denote the upper bound of the distance gap between the optimal location and prediction location, i.e., $\eta = \sup |\hat{y} - y^*|$, and $\rho_\alpha(\eta)$ be the approximation ratio for any specific $\alpha$ under prediction error $\eta$.

**Theorem 3.4.** *Let $\eta$ denote $\sup |\hat{y} - y^*|$. When $\alpha \in [1, \frac{1+\sqrt{5}}{2}]$, the approximation ratio is*

$$
\rho_\alpha(\eta) = \begin{cases} \alpha & \eta \in [0, \frac{\alpha-1}{2(\alpha+1)}] \\ 1 + \frac{4\eta}{1-2\eta} & \eta \in (\frac{\alpha-1}{2(\alpha+1)}, \frac{1}{\alpha} - \frac{1}{2}] \\ 1 + \frac{2\alpha\eta}{\alpha-1} & \eta \in (\frac{1}{\alpha} - \frac{1}{2}, \frac{1}{2\alpha}] \\ \frac{\alpha}{\alpha-1} & \eta \in (\frac{1}{2\alpha}, +\infty) \end{cases} .
$$

*When $\alpha \in (\frac{1+\sqrt{5}}{2}, 2]$, the approximation ratio is*

$$
\rho_\alpha(\eta) = \begin{cases} \alpha & \eta \in [0, \frac{(\alpha-1)^2}{2\alpha}] \\ 1 + \frac{2\alpha\eta}{\alpha-1} & \eta \in (\frac{(\alpha-1)^2}{2\alpha}, \frac{1}{2\alpha}] \\ \frac{\alpha}{\alpha-1} & \eta \in (\frac{1}{2\alpha}, +\infty) \end{cases} .
$$

*Proof Sketch.*   Recall that $\alpha$-BIM places the facility at $\hat{y}$ when it is in the interval $[1 - \frac{1}{\alpha}, \frac{1}{\alpha}]$, and otherwise places it at the nearest endpoint of that interval. We obtain the approximation ratio as a function of $\eta$ by treating these two placement regimes separately and taking the worst case in each. When $\hat{y} \in [1 - \frac{1}{\alpha}, \frac{1}{\alpha}]$, by Lemma 3.1, we focus on the two-agent instances and analyze the worst case approximation ratio parameterized by $\eta$ when moving the facility from $y^*$ to $\hat{y}$ as $\eta$ grows from $0$ to $\infty$. Similarly, when $\hat{y} \notin [1 - \frac{1}{\alpha}, \frac{1}{\alpha}]$ and the facility is placed at the nearest endpoint. We again consider the worst case when moving the facility from $y^*$ to $1 - \frac{1}{\alpha}$ (or $\frac{1}{\alpha}$). Taking the worst case over the two placement regimes produces a single approximation ratio expressed as a piecewise function of $\eta$, which monotonically increases from $\alpha$ to $\frac{\alpha}{\alpha-1}$ as the error bound $\eta$ increases continuously.   □

To better illustrate Theorem 3.4, we present the approximation ratios in Figure 2. For each fixed value of $\alpha$, the approximation ratio is a piecewise function, which is smooth, specifically, continuous and monotonic with respect to error bound $\eta$. We further observe that when the error bound $\eta \leq \frac{\sqrt{5}}{2} - 1 \approx 0.118$, the approximation ratio achieves $\frac{\sqrt{5}+1}{2} \approx 1.62$, which shows that $\alpha$-BIM can substantially improve the ratio with a well-performed prediction model.

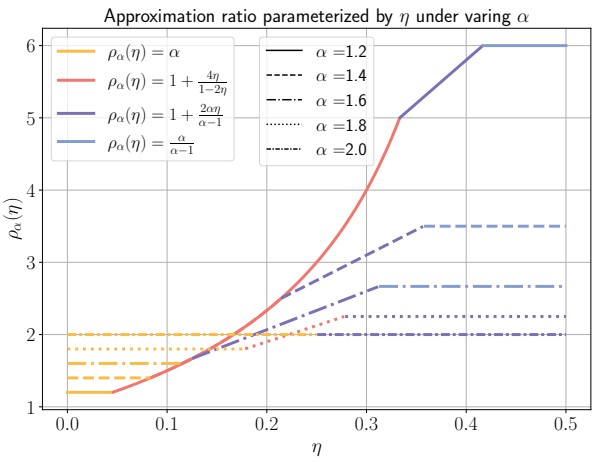

Figure 2: Approximation ratio parameterized by error bound $\eta$ with various $\alpha$ values.

## 4   Randomized Mechanisms

In the context of randomized mechanism design (without predictions) for envy ratio minimization, Ding et al. [2020] proved that any strategyproof mechanism must have an approximation ratio of at

least 1.0314, and showed that an approximation ratio of 2 is achieved by the deterministic mechanism which always places the facility at $\frac{1}{2}$. However, they were unable to construct any randomized strategyproof mechanism beyond 2-approximation. In this section, we address this gap and open problem within both the classic setting (without predictions), and mechanism design with predictions.

## 4.1 Without Prediction

Since it remains an open question whether a randomized mechanism can achieve an approximation ratio better than 2, we address this by introducing a novel family of $(\alpha, p)$-LRM constant mechanisms (Algorithm 2). The mechanism is inherently strategyproof and anonymous. By carefully selecting the parameters, we show that there exists a mechanism within this family that achieves an approximation ratio of approximately 1.8944.

---

**Mechanism 2** $(\alpha, p)$-LRM Constant Mechanism

---

**Input:** Location profile $\mathbf{x}$.
**Output:** Distribution of facility locations $f(\mathbf{x})$.
  1: With probability $p$: return $f(\mathbf{x}) = \frac{1}{2} - \alpha$;
  2: With probability $1 - 2p$: return $f(\mathbf{x}) = \frac{1}{2}$;
  3: With probability $p$: return $f(\mathbf{x}) = \frac{1}{2} + \alpha$;

---

We now compute the optimal parameters of $\alpha$ and $p$ which minimize the mechanism's approximation ratio. By the following lemma, we show that it suffices to restrict our attention to mechanisms with $\alpha \leq \frac{1}{4}$, as any $(\alpha, p)$-LRM constant mechanism with $\alpha > \frac{1}{4}$ will have a worse approximation ratio than the deterministic mechanism which simply places the facility at $\frac{1}{2}$.

**Lemma 4.1.** *When $\alpha > \frac{1}{4}$, every $(\alpha, p)$-LRM constant mechanism has an approximation ratio of at least 2.*

Given that $\alpha \leq \frac{1}{4}$, we show that the optimal parameters of $\alpha$ and $p$ can be found by solving the following optimization problem, which concerns the mechanism's performance over 2 different location profiles.

**Lemma 4.2.** *Let $\mathbf{x} = (x_1 = 0, x_2 = \frac{1}{2})$ and $\mathbf{x}' = (x'_1 = 0, x'_2 = \frac{1}{2} + \alpha)$. When $\alpha \leq \frac{1}{4}$, the $(\alpha^*, p^*)$-LRM constant Mechanism optimizes the approximation ratio of envy ratio objective where $(\alpha^*, p^*) = \arg\min_{(\alpha, p)} \{\max\{\rho(\mathbf{x}), \rho(\mathbf{x}')\}\}$.*

Finally, by solving this optimization problem, we show that setting $\alpha = \frac{\sqrt{5}}{2} - 1$ and $p = \frac{2}{5}$ leads to the optimal approximation ratio among all $(\alpha, p)$-LRM constant mechanisms.

**Theorem 4.3.** *$(\frac{\sqrt{5}}{2} - 1, \frac{2}{5})$-LRM constant mechanism is anonymous, strategyproof, and achieves an approximation ratio of $1 + \frac{2}{\sqrt{5}}$, which is optimal among all $(\alpha, p)$-LRM constant mechanisms.*

*Proof Sketch of Theorem 4.3.* By Lemma 4.2, it suffices to find the optimal parameters $\alpha^*$ and $p^*$ for the optimization problem $\min_{(\alpha, p)} \{\max\{\rho(\mathbf{x}), \rho(\mathbf{x}')\}\}$. Specifically, we show that when $\alpha \in [0, \frac{1}{6})$, by setting $\alpha = \frac{\sqrt{5}}{2} - 1$, and $p = \frac{2}{5}$, the $(\frac{\sqrt{5}}{2} - 1, \frac{2}{5})$-LRM mechanism achieves an approximation ratio of $1 + \frac{2}{\sqrt{5}} \approx 1.8944$ while when $\alpha \in [\frac{1}{6}, \frac{1}{4}]$, the optimal parameters are $\alpha = \frac{1}{6}$, and $p = \frac{4}{11}$, which yields an approximation ratio of $\frac{21}{11} \approx 1.909$. Therefore, the $(\frac{\sqrt{5}}{2} - 1, \frac{2}{5})$-LRM mechanism is optimal within the family of $(\alpha, p)$-LRM Constant mechanisms. $\square$

With an approximation ratio of approximately 1.8944, our $(\frac{\sqrt{5}}{2} - 1, \frac{2}{5})$-LRM Constant mechanism significantly improves upon the upper bound among mechanisms without predictions. We also further tighten the gap by establishing an improved lower bound. Previously, Ding et al. [2020] showed that any randomized strategyproof mechanism (without predictions) has an approximation ratio of at least 1.0314. We advance this lower bound by carefully selecting a location profile and constructing an upper bound on the facility's expected distance from an agent's location, in terms of its probability to be located within certain intervals, which gives us a lower bound of 1.12579.

**Theorem 4.4.** *Any randomized strategyproof mechanism has an approximation ratio of at least 1.12579.*

## 4.2 With Prediction

Next, we extend our investigation to the paradigm of randomized mechanism design with predictions, proposing a new randomized mechanism that outperforms the $\alpha$-BIM. An immediate idea may be to run the $(\frac{\sqrt{5}}{2} - 1, \frac{2}{5})$-LRM constant mechanism within the $\alpha$-BIM, returning the former mechanism's output when the prediction lies outside the bounding interval. However, this modification performs worse than the original $\alpha$-BIM. Further details are provided in Appendix C.

To demonstrate the difficulty of this problem, consider an extreme 2-agent instance $\mathbf{x} = (x_1 = 0, x_2 = 1)$, with prediction $\hat{y} = 0$. For any mechanism that places the facility at $\hat{y}$ with positive probability, the robustness of the mechanism becomes unbounded. However, intuitively, assigning a higher probability to placing the facility at $\hat{y}$ improves consistency. This reveals the fundamental challenge of balancing consistency and robustness. To address this, we adapt the underlying design principle of the $\alpha$-BIM: the mechanism locates the facility at $\hat{y}$ if $\hat{y}$ lies within a specified closed interval. Otherwise, the facility is placed at the boundary of that interval. This design can be viewed as a threshold mechanism, where the placement decision is based on the distance between $\hat{y}$ and $\frac{1}{2}$. By integrating this threshold-based approach in a probabilistic manner, we design our novel Bias-Aware mechanism, which we introduce as follows.

---

**Mechanism 3** Bias-Aware Mechanism (BAM)

---

**Input:** Location profile $\mathbf{x}$, facility location prediction $\hat{y}$.
**Output:** Facility location $f(\mathbf{x}, \hat{y})$.
1: Compute bias $c = |\hat{y} - \frac{1}{2}|$
2: Compute probability $p = \frac{1}{2} - c$
3: With probability $p$: return $f(\mathbf{x}, \hat{y}) = \hat{y}$
4: With probability $1 - p$: return $f(\mathbf{x}, \hat{y}) = \frac{1}{2}$

---

**Theorem 4.5.** *BAM is anonymous, strategyproof and $(-4c^2 + 2)$-consistency and $(c + 2)$-robustness when $c \in [\frac{1}{4}, \frac{1}{2}]$, $\frac{7}{4}$-consistency and $\frac{9}{4}$-robustness when $c \in [0, \frac{1}{4})$.*

*Proof.* BAM is immediately anonymous and strategyproof as the output is independent of the agents' locations. We now consider its consistency and robustness. From Lemma 3.1, we only need to consider instances with two agents, in which the optimal envy ratio is always 1. Hence, we only need to consider the envy ratio achieved by the mechanism. Given any profile $\mathbf{x}$ and $\hat{y}$, without loss of generality, we assume that $x_1 < x_2$ and $\hat{y} \leq \frac{1}{2}$, giving us $p = \hat{y}$ and $c = \frac{1}{2} - \hat{y}$.

**(Robustness).** We first consider robustness. Observe that when placing the facility at $\hat{y}$ (resp. $\frac{1}{2}$), the minimum utility of any agent is at least $1 - \hat{y}$ (resp. $\frac{1}{2}$) as the distance from $\hat{y}$ (resp. $\frac{1}{2}$) is at most $1 - \hat{y}$ (resp. $\frac{1}{2}$). The equalities hold when $x_2 = 1$. If $x_1 < \hat{y}$, moving $x_1$ to $\hat{y}$ will increase the maximum utility achieved by $\hat{y}$ and $\frac{1}{2}$. If $x_1 > \frac{1}{2}$, moving $x_1$ to $\frac{1}{2}$ will increase the maximum utility achieved by $\hat{y}$ and $\frac{1}{2}$. Hence, we only need to consider the case where $x_1 \in [\hat{y}, \frac{1}{2}]$, in which the robustness is expressed as

$$\mathrm{ER}(f(\mathbf{x}, \hat{y}), \mathbf{x}) = \hat{y} \cdot \frac{1 - (x_1 - \hat{y})}{\hat{y}} + (1 - \hat{y}) \cdot \frac{1 - (\frac{1}{2} - x_1)}{\frac{1}{2}} \leq \frac{5}{2} - \hat{y} = 2 + c,$$

which reaches the maximum when $x_1$ reaches $\frac{1}{2}$.

**(Consistency).** Consider any arbitrary instance $\mathbf{x}$ and $\hat{y}$ is accurate, i.e., $\hat{y} = \mathrm{mid}(\mathbf{x})$. Let $\delta = \frac{x_2 - x_1}{2}$. If $\hat{y} \leq \frac{1}{4}$, we have $x_2 \leq \frac{1}{2}$. The envy ratio achieved by $f(\mathbf{x}, \hat{y}) = \hat{y}$ is 1 and the envy ratio achieved by $f(\mathbf{x}, \hat{y}) = \frac{1}{2}$ is $\frac{1 - (\frac{1}{2} - x_2)}{1 - (\frac{1}{2} - x_1)} = \frac{1 - (\frac{1}{2} - \hat{y} - \delta)}{1 - (\frac{1}{2} - \hat{y} + \delta)}$, where we have $\delta \leq \hat{y}$ as $0 \leq x_1 \leq \hat{y}$. For consistency, it is

$$\mathrm{ER}(f(\mathbf{x}, \hat{y}), \mathbf{x}) = \hat{y} \cdot 1 + (1 - \hat{y}) \cdot \frac{1 - (\frac{1}{2} - \hat{y} - \delta)}{1 - (\frac{1}{2} - \hat{y} + \delta)} \leq \hat{y} + (1 - \hat{y})(1 + 4\hat{y}) = -4c^2 + 2,$$

which reaches the maximum when $\delta = \hat{y}$.

When $\hat{y} > \frac{1}{4}$, the envy ratio achieved by $f(\mathbf{x}, \hat{y}) = \hat{y}$ is 1 and the envy ratio achieved by $f(\mathbf{x}, \hat{y}) = \frac{1}{2}$ is $\frac{1-(\delta-(\frac{1}{2}-\hat{y}))}{1-(\delta+\frac{1}{2}-\hat{y})}$, where $\delta \leq \hat{y}$ as $0 \leq x_1 \leq \hat{y}$. Hence, the consistency is

$$\mathrm{ER}(f(\mathbf{x}, \hat{y}), \mathbf{x}) = \hat{y} \cdot 1 + (1-\hat{y}) \frac{1-(\delta-(\frac{1}{2}-\hat{y}))}{1-(\delta+\frac{1}{2}-\hat{y})} \leq \hat{y} + (1-\hat{y})(3-4\hat{y}) = 4c^2 + 2c + 1,$$

which is maximized when $\delta = \hat{y}$. Note that in this case, both consistency and robustness are monotonically increasing w.r.t. $c$, reaching the maximum of $(\frac{7}{4}, \frac{9}{4})$ when $c = \frac{1}{4}$.

Finally, we conclude that BAM satisfies $(-4c^2 + 2)$-consistency and $(c + 2)$-robustness when $c \in [\frac{1}{4}, \frac{1}{2}]$, $\frac{7}{4}$-consistency and $\frac{9}{4}$-robustness when $c \in [0, \frac{1}{4})$. $\qquad\square$

Intuitively, BAM reduces the probability that the facility is placed at $\hat{y}$ as the distance between $\hat{y}$ and the midpoint increases, in which the probability reaches 0 when $\hat{y}$ reaches 0 or 1. This prevents the mechanism from having unbounded robustness, and improves the balance between consistency and robustness by effectively using the prediction. Figure 3 highlights that BAM is strictly better than the deterministic $\alpha$-BIM in terms of both consistency and robustness. Further discussion is provided in Appendix C. For instance, we show that if we modify BAM by replacing the $\frac{1}{2}$ output with the $(\frac{\sqrt{5}}{2} - 1, \frac{2}{5})$-LRM, both the consistency and robustness worsen.

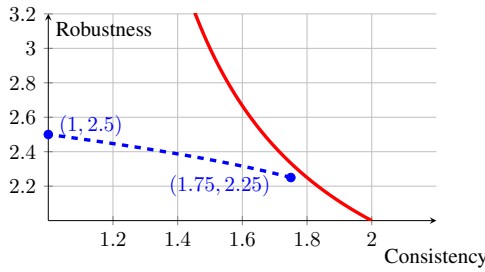

Figure 3: Comparison between $\alpha$-BIM (red solid line) and BAM (blue dashed line). Note that, unlike $\alpha$-BIM, the range of approximation ratios for BAM is not dependent on a chosen parameter, but rather on $|\hat{y} - \frac{1}{2}|$.

## 5 Conclusion and Discussion

In this paper, we revisit the problem of facility location mechanism design problems for the envy ratio objective, through the scope of learning-augmentation. We provide tight results by devising the deterministic $\alpha$-BIM, which reaches the Pareto frontier of deterministic, anonymous and strategyproof mechanisms. For randomized mechanisms without prediction, we improve upon the best-known lower bound, and propose the $(\frac{\sqrt{5}}{2} - 1, \frac{2}{5})$-LRM Constant mechanism which achieves a 1.8944-approximation, resolving the open question of devising a mechanism with an approximation ratio better than 2. Finally, we proposed BAM, a learning-augmented randomized mechanism which outperforms $\alpha$-BIM in terms of both consistency and robustness. For $\alpha$-BIM , we provide a comprehensive analysis regarding the approximation ratio parameterized by prediction error, however, regarding BAM, the randomized mechanism with prediction, unfortunately, we are unable to derive a closed-form expression for approximation ratio $\rho(\eta)$. The inherent difficulty is that when one performs a case-by-case analysis, each case is expressed in the form $\max_j \{f_j(\eta, \hat{y})\}$, in which the presence of $\hat{y}$ in the probability terms introduces significant complexity, especially the numerator, which includes quadratic expressions and cross terms. This prevents us from deriving an explicit form for the $\eta$-parameterized approximation ratio.

For future work, exploring lower bounds for randomized mechanisms with predictions presents a challenging yet meaningful task. Due to the significant differences in optimization objectives, the state-of-the-art techniques used by Balkanski et al. [2024] for learning-augmented randomized mechanisms are difficult to extend to the envy ratio scenario studied in this paper. Thus, developing a novel approach to establish lower bounds would be beneficial. Additionally, it is promising to apply the learning-augmented framework to other fairness notions within the literature.

## Acknowledgments

This work was supported by the NSF-CSIRO grant on "Fair Sequential Collective Decision-Making" (RG230833) and the ARC Laureate Project FL200100204 on "Trustworthy AI". The authors would like to express their gratitude to the anonymous reviewers of IJCAI 2025 and NeurIPS 2025 for their insightful and constructive feedback, which greatly helped improve this paper.

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

# A   Omitted Proofs for Section 3

## A.1   Proof of Lemma 3.1

*Proof.* For an arbitrary location profile $\mathbf{x} = (x_1, \ldots, x_n)$, without loss of generality, we assume that $x_1 \leq \ldots \leq x_n$. Let $\tilde{u}$ and $\bar{u}$ denote the maximum and minimum agent utilities, respectively, when the facility is placed at $\mathrm{mid}(\mathbf{x})$. Consequently, the optimal envy ratio can be expressed as $\mathrm{ER}(\mathrm{mid}(\mathbf{x}), \mathbf{x}) = \frac{\tilde{u}}{\bar{u}}$.

For any facility location $y \in P$, if $y \in [x_1, x_n]$, the maximum utility is at most $\tilde{u} + |y - \mathrm{mid}(\mathbf{x})|$, while the minimum utility under $y$ is at least $\bar{u} - |y - \mathrm{mid}(\mathbf{x})|$. Therefore, we derive the following inequality:

$$\frac{\mathrm{ER}(y, \mathbf{x})}{\mathrm{ER}(\mathrm{mid}(\mathbf{x}), \mathbf{x})} \leq \frac{(\tilde{u} + |y - \mathrm{mid}(\mathbf{x})|)/(\bar{u} - |y - \mathrm{mid}(\mathbf{x})|)}{\tilde{u}/\bar{u}}$$

$$\leq \frac{(\bar{u} + |y - \mathrm{mid}(\mathbf{x})|)/(\bar{u} - |y - \mathrm{mid}(\mathbf{x})|)}{\bar{u}/\bar{u}} \quad (\because \text{ratio is non-increasing w.r.t. } \tilde{u})$$

$$= \frac{(\bar{u} + |y - \mathrm{mid}(\mathbf{x})|)}{(\bar{u} - |y - \mathrm{mid}(\mathbf{x})|)},$$

Equality holds when all the agents are at $x_1$ and $x_n$. In this case, consider a new 2-agent instance $\mathbf{x}' = (x_1, x_n)$. Notice that both $x_1$ and $x_n$ achieve the minimum utility under $\mathrm{mid}(\mathbf{x})$, i.e., $u(\mathrm{mid}(\mathbf{x}), x_1) = u(\mathrm{mid}(\mathbf{x}), x_n) = \bar{u}$. For this instance, the approximation ratio of placing the facility at $y$ is

$$\frac{\mathrm{ER}(y, \mathbf{x}')}{\mathrm{ER}(\mathrm{mid}(\mathbf{x}'), \mathbf{x}')} = \frac{\bar{u} + |y - \mathrm{mid}(\mathbf{x}')|/\bar{u} - |y - \mathrm{mid}(\mathbf{x}')|}{1}$$

$$= \frac{(\bar{u} + |y - \mathrm{mid}(\mathbf{x})|)}{(\bar{u} - |y - \mathrm{mid}(\mathbf{x})|)} \geq \frac{\mathrm{ER}(y, \mathbf{x})}{\mathrm{ER}(\mathrm{mid}(\mathbf{x}), \mathbf{x})}.$$

For the case where $y \notin [x_1, x_n]$, without loss of generality, assume $y > x_n$. Since $x_1$ and $x_n$ always achieve the minimum utility under $\mathrm{mid}(\mathbf{x})$, we have $d(\mathrm{mid}(\mathbf{x}), x_1) = d(\mathrm{mid}(\mathbf{x}), x_n) = 1 - \bar{u}$. When changing the facility location to $y > x_n$, $x_1$ achieves the minimum utility while $x_n$ achieves the maximum utility. Specifically, $u(y, x_1) = 1 - d(y, x_1) = 1 - (y - \mathrm{mid}(\mathbf{x}) + (1 - \bar{u})) = -y + \mathrm{mid}(\mathbf{x}) + \bar{u}$, while $u(y, x_n) = 1 - d(y, x_n) = 1 - (y - \mathrm{mid}(\mathbf{x}) - (1 - \bar{u})) = 2 - y + \mathrm{mid}(\mathbf{x}) - \bar{u}$. Thus, the envy ratio is

$$\frac{\mathrm{ER}(y, \mathbf{x})}{\mathrm{ER}(\mathrm{mid}(\mathbf{x}), \mathbf{x})} \leq \frac{(2 - y + \mathrm{mid}(\mathbf{x}) - \bar{u})/(-y + \mathrm{mid}(\mathbf{x}) + \bar{u})}{\tilde{u}/\bar{u}}$$

$$\leq \frac{2 - y + \mathrm{mid}(\mathbf{x}) - \bar{u}}{-y + \mathrm{mid}(\mathbf{x}) + \bar{u}}. \quad (\because \text{ratio is non-increasing w.r.t. } \tilde{u})$$

Equality holds when all the agents are located at $x_1$ and $x_n$. By considering the new instance $\mathbf{x}' = (x_1, x_n)$, it can be verified that

$$\frac{\mathrm{ER}(y, \mathbf{x}')}{\mathrm{ER}(\mathrm{mid}(\mathbf{x}'), \mathbf{x}')} = \frac{2 - y + \mathrm{mid}(\mathbf{x}') - \bar{u}}{-y + \mathrm{mid}(\mathbf{x}') + \bar{u}}$$

$$= \frac{2 - y + \mathrm{mid}(\mathbf{x}) - \bar{u}}{-y + \mathrm{mid}(\mathbf{x}) + \bar{u}} \geq \frac{\mathrm{ER}(y, \mathbf{x})}{\mathrm{ER}(\mathrm{mid}(\mathbf{x}), \mathbf{x})}.$$

In conclusion, for any location profile $\mathbf{x}$ and distribution of facility locations $P$, we can always construct a new 2-agent instance $\mathbf{x}' = (\mathrm{lm}(\mathbf{x}), \mathrm{rm}(\mathbf{x}))$ such that the approximation ratio of any $y \in P$ under $\mathbf{x}$ is upper-bounded by the approximation ratio of $y$ under $\mathbf{x}'$. Formally,

$$\mathbb{E}_{y \in P}\left[\frac{\mathrm{ER}(y, \mathbf{x})}{\mathrm{ER}(\mathrm{mid}(\mathbf{x}), \mathbf{x})}\right] \leq \mathbb{E}_{y \in P}\left[\frac{\mathrm{ER}(y, \mathbf{x}')}{\mathrm{ER}(\mathrm{mid}(\mathbf{x}'), \mathbf{x}')}\right].$$

This completes the proof.   □

## A.2 Proof of Theorem 3.4

*Proof.* When $\hat{y} \in [1 - \frac{1}{\alpha}, \frac{1}{\alpha}]$, the $\alpha$-BIM mechanism places the facility at $\hat{y}$. Without loss of generality, assume that $y^* \in [0, \frac{1}{2}]$ (the case $y^* \in [\frac{1}{2}, 1]$ is symmetric). Suppose the prediction error satisfies $|\hat{y} - y^*| \leq \eta$. We analyze the approximation ratio by cases on $\eta$. Note that for $\alpha \in [1, 2]$, it always holds that $\frac{1}{\alpha} - \frac{1}{2} \leq \frac{1}{2\alpha}$.

**Case 1.** $\eta \in [0, \frac{1}{\alpha} - \frac{1}{2}]$. By Lemma 3.1, we may restrict to a two-agent instance $\mathbf{x} = (x_1, x_2)$ with $x_2 = 2y^* - x_1$. Since $y^*$ is optimal, the approximation ratio for placing the facility at $\hat{y}$ is bounded by

$$
\begin{aligned}
\rho &\leq \frac{1 - (y^* - x_1) + \eta}{1 - (y^* - x_1) - \eta} \\
&\leq \frac{1 - \frac{1}{2} + \eta}{1 - \frac{1}{2} - \eta} \qquad\qquad \text{(since } x_1 \geq 0, \ y^* \leq \frac{1}{2}) \\
&= 1 + \frac{4\eta}{1 - 2\eta}.
\end{aligned}
$$

**Case 2.** $\eta \in (\frac{1}{\alpha} - \frac{1}{2}, \frac{1}{2\alpha}]$. Here the prediction error exceeds $\frac{1}{\alpha} - \frac{1}{2}$, implying a tighter bound on $y^*$ since $y^* \leq \frac{1}{2}$ and $\hat{y} \in [1 - \frac{1}{\alpha}, \frac{1}{\alpha}]$. Thus, $y^* \leq \hat{y} - \eta \leq \frac{1}{\alpha} - \eta \leq \frac{1}{2}$.

Consider any instance $\mathbf{x} = (x_1, x_2 = 2y^* - x_1)$. When the facility moves from $y^*$ to $\hat{y}$, the maximum utility can increase by at most $\eta$, and the minimum utility can decrease by at most $\eta$. Hence,

$$
\begin{aligned}
\rho &\leq \frac{1 - (y^* - x_1) + \eta}{1 - (y^* - x_1) - \eta} \\
&\leq \frac{1 - (\frac{1}{\alpha} - \eta) + \eta}{1 - (\frac{1}{\alpha} - \eta) - \eta} \qquad\qquad \text{(since } x_1 \geq 0, \ y^* \leq \frac{1}{\alpha} - \eta) \\
&= 1 + \frac{2\alpha\eta}{\alpha - 1}.
\end{aligned}
$$

**Case 3.** $\eta > \frac{1}{2\alpha}$. In this regime, the approximation ratio is upper-bounded by the robustness ratio: $\rho \leq \frac{1}{1 - \frac{1}{\alpha}} = \frac{\alpha}{\alpha - 1}$. This bound is tight for the instance $\mathbf{x} = (0, \frac{1}{\alpha})$, where $y^* = \frac{1}{2\alpha}$ and $\hat{y} = \frac{1}{\alpha}$.

Next, consider $\hat{y} \in [0, 1 - \frac{1}{\alpha})$ or $\hat{y} \in (\frac{1}{\alpha}, 1]$, where $\alpha$-BIM places the facility at the endpoint $1 - \frac{1}{\alpha}$ or $\frac{1}{\alpha}$. Without loss of generality, we analyze the case $\hat{y} = 1 - \frac{1}{\alpha}$.

**Case 1.** $y^* \leq 1 - \frac{1}{\alpha}$. Here $x_2$ is closer to $\hat{y}$ than $x_1$ is. The utility of agent 1 is at least $\frac{1}{\alpha}$, and that of agent 2 is at most 1, yielding $\rho \leq \frac{1}{1/\alpha} = \alpha$. Equality holds when $\hat{y} = y^* = \frac{1}{2} - \frac{1}{2\alpha}$ for any $\eta \geq 0$.

**Case 2.** $y^* > 1 - \frac{1}{\alpha}$. We further distinguish subcases:

(a) If $\eta \in [0, \frac{1}{\alpha} - \frac{1}{2}]$, then $y^* = \hat{y} + \eta \leq 1 - \frac{1}{\alpha} + \eta \leq \frac{1}{2}$. Hence,

$$
\rho \leq \frac{\frac{1}{\alpha}}{1 - (2y^* - (1 - \frac{1}{\alpha}))} \leq \frac{\frac{1}{\alpha}}{1 - (2(1 - \frac{1}{\alpha} + \eta) - (1 - \frac{1}{\alpha}))} = \frac{1}{1 - 2\alpha\eta}.
$$

(b) If $\eta \in (\frac{1}{\alpha} - \frac{1}{2}, \frac{1}{2\alpha}]$, for any instance $\mathbf{x} = (x_1, 2y^* - x_1)$, moving $x_1$ to $1 - \frac{1}{\alpha}$ (if it lies to the right) increases the ratio while maintaining $y^* - \hat{y} \leq \eta$. Thus, we only need to consider $x_1 \leq 1 - \frac{1}{\alpha}$, giving

$$
\begin{aligned}
\rho &\leq \frac{1 - ((1 - \frac{1}{\alpha}) - x_1)}{1 - ((2y^* - x_1) - (1 - \frac{1}{\alpha}))} \\
&\leq \frac{\frac{1}{\alpha} + 2y^* - 1}{1 - \frac{1}{\alpha}} \leq \frac{1 - \frac{1}{\alpha} + 2\eta}{1 - \frac{1}{\alpha}} = 1 + \frac{2\alpha\eta}{\alpha - 1},
\end{aligned}
$$

where the second inequality follows from $x_1 \geq 2y^* - 1$, and the third from $y^* \leq 1 - \frac{1}{\alpha} + \eta$.

(c) If $\eta > \frac{1}{2\alpha}$, the approximation ratio is bounded by $\frac{\alpha}{\alpha-1}$, which is tight for $\mathbf{x} = (1 - \frac{1}{\alpha}, 1)$ with $y^* = 1 - \frac{1}{2\alpha}$ and $\hat{y} = 1 - \frac{1}{\alpha} - \varepsilon$ for any $\varepsilon > 0$.

Combining the above, when $\hat{y} \in [0, 1 - \frac{1}{\alpha})$ or $(\frac{1}{\alpha}, 1]$, we obtain:

$$\rho \leq \begin{cases} \alpha, & \eta \in [0, \frac{\alpha-1}{2\alpha^2}], \\ \frac{1}{1-2\alpha\eta}, & \eta \in [\frac{\alpha-1}{2\alpha^2}, \frac{1}{\alpha} - \frac{1}{2}], \\ 1 + \frac{2\alpha\eta}{\alpha-1}, & \eta \in [\frac{1}{\alpha} - \frac{1}{2}, \frac{1}{2\alpha}], \\ \frac{\alpha}{\alpha-1}, & \eta \in [\frac{1}{2\alpha}, +\infty). \end{cases}$$

Finally, we combine all bounds by distinguishing two parameter ranges.

- When $\alpha \in [1, \frac{1+\sqrt{5}}{2}]$, note that $\frac{\alpha-1}{2(\alpha+1)} \leq \frac{1}{\alpha} - \frac{1}{2}$ and that $\alpha \geq \frac{1+2\eta}{1-2\eta}$ for $\eta \leq \frac{\alpha-1}{2(\alpha+1)}$. Hence:

$$\rho_\alpha(\eta) = \begin{cases} \alpha, & \eta \in [0, \frac{\alpha-1}{2(\alpha+1)}], \\ 1 + \frac{4\eta}{1-2\eta}, & \eta \in (\frac{\alpha-1}{2(\alpha+1)}, \frac{1}{\alpha} - \frac{1}{2}], \\ 1 + \frac{2\alpha\eta}{\alpha-1}, & \eta \in (\frac{1}{\alpha} - \frac{1}{2}, \frac{1}{2\alpha}], \\ \frac{\alpha}{\alpha-1}, & \eta \in (\frac{1}{2\alpha}, +\infty). \end{cases}$$

- When $\alpha \in (\frac{1+\sqrt{5}}{2}, 2]$, we obtain:

$$\rho_\alpha(\eta) = \begin{cases} \alpha, & \eta \in [0, \frac{(\alpha-1)^2}{2\alpha}], \\ 1 + \frac{2\alpha\eta}{\alpha-1}, & \eta \in (\frac{(\alpha-1)^2}{2\alpha}, \frac{1}{2\alpha}], \\ \frac{\alpha}{\alpha-1}, & \eta \in (\frac{1}{2\alpha}, +\infty). \end{cases}$$

This completes the analysis of approximation ratio parameterized by prediction error bound. □

# B  Omitted Proofs for Section 4

## B.1  Proof of Lemma 4.1

*Proof.* For any arbitrary $(\alpha, p)$-LRM constant mechanism with $\alpha \in (\frac{1}{4}, \frac{1}{2}]$, we first consider a 2-agent instance $\mathbf{x} = (x_1 = 0, x_2 = \frac{1}{2})$. The approximation ratio $\rho(\mathbf{x})$ under the instance $\mathbf{x}$ can be represented as

$$\rho(\mathbf{x}) = p \cdot \frac{1-\alpha}{1 - (\frac{1}{2} - \alpha)} + (1 - 2p) \cdot \frac{1-0}{1 - \frac{1}{2}} + p \cdot \frac{1-\alpha}{1 - (\frac{1}{2} + \alpha)}$$

$$= 2(1 - 2p) + p \cdot \frac{4 - 4\alpha}{1 - 4\alpha^2}.$$

It is straightforward to verify that $\rho(\mathbf{x})$ is monotonically increasing with respect to $\alpha$ for $\alpha \in (\frac{1}{4}, \frac{1}{2}]$. This implies that $\rho(\mathbf{x}) \geq 2(1 - 2p) + p \cdot \frac{4-1}{1-\frac{1}{4}} = 2(1 - 2p) + 4p = 2$ with equality attained when $\alpha = \frac{1}{4}$. In other words, as long as $\alpha > \frac{1}{4}$, the approximation of $(\alpha, p)$-LRM is at least 2, regardless of the choice of parameter $p \in [0, \frac{1}{2}]$. □

## B.2  Proof of Lemma 4.2

*Proof.* Note that by Lemma 3.1, it suffices to consider two-agent instances $\mathbf{x} = (x_1, x_2)$. Without loss of generality, we assume $0 \leq x_1 \leq x_2 \leq 1$. Let $\text{mid}(\mathbf{x}) = \frac{x_1 + x_2}{2}$ be the midpoint of the agents' locations. By symmetry, we focus on the case where $\text{mid}(\mathbf{x}) \in [0, \frac{1}{2}]$ (the analysis for $\text{mid}(\mathbf{x}) \in [\frac{1}{2}, 1]$ follows analogously). Let $u$ denote the utility of both agents when the facility is

placed at $\mathrm{mid}(\mathbf{x})$, and define $\delta = \frac{1}{2} - \mathrm{mid}(\mathbf{x})$ as the distance between $\mathrm{mid}(\mathbf{x})$ and $\frac{1}{2}$. Consequently, we derive that $u \geq \frac{1}{2} + \delta$ as $u = 1 - (\mathrm{mid}(\mathbf{x}) - x_1) \geq 1 - \mathrm{mid}(\mathbf{x}) = 1 - (\frac{1}{2} - \delta) = \frac{1}{2} + \delta$. We first consider the situation where $\alpha \in [0, \frac{1}{6})$, that is $1 - 2\alpha > \frac{1}{2} + \alpha$. We consider the following two cases.

**Case (1).** $\delta \in [0, \alpha]$, i.e., $\mathrm{mid}(\mathbf{x}) \in [\frac{1}{2} - \alpha, \frac{1}{2}]$. For any such instance $\mathbf{x} = (x_1, x_2)$, the approximation ratio is upper-bounded by

$$\rho(\mathbf{x}) \leq p \cdot \frac{u + (\alpha - \delta)}{u - (\alpha - \delta)} + (1 - 2p) \cdot \frac{u + \delta}{u - \delta} + p \cdot \frac{u + (\alpha + \delta)}{u - (\alpha + \delta)}.$$

This bound follows from the observation that moving the facility location from $\mathrm{mid}(\mathbf{x})$ to $\frac{1}{2} - \alpha$ results in a maximum utility increase (decrease) of $\alpha - \delta$, regardless of whether $x_1$ or $x_2$ is closer to $\mathrm{mid}(\mathbf{x})$. Similarly, moving the facility to $\frac{1}{2}$ or $\frac{1}{2} + \alpha$ changes an agent's utility by at most $\delta$ or $\alpha + \delta$, respectively.

Notice that $\rho(\mathbf{x})$ is monotonically non-increasing with respect to $u$ and $u \geq \frac{1}{2} + \delta$, we further upper-bound the ratio by

$$\begin{aligned}
\rho(\mathbf{x}) &\leq p \cdot \frac{\frac{1}{2} + \alpha}{\frac{1}{2} + 2\delta - \alpha} + (1 - 2p) \cdot \frac{\frac{1}{2} + 2\delta}{\frac{1}{2}} + p \cdot \frac{\frac{1}{2} + 2\delta + \alpha}{\frac{1}{2} - \alpha} \\
&= p \cdot \frac{1 + 2\alpha}{1 + 4\delta - 2\alpha} + (1 - 2p) \cdot (1 + 4\delta) + p \cdot \frac{1 + 4\delta + 2\alpha}{1 - 2\alpha} \\
&\leq p \cdot \frac{1 + 2\alpha - 4\delta}{1 - 2\alpha} + (1 - 2p) \cdot (1 + 4\delta) + p \cdot \frac{1 + 4\delta + 2\alpha}{1 - 2\alpha} \\
&\qquad\qquad\qquad (\because \text{convexity of } \tfrac{1+2\alpha}{1+4\delta-2\alpha} \text{ and } 0 \leq \delta \leq \alpha) \\
&= 2p \cdot \frac{1 + 2\alpha}{1 - 2\alpha} + (1 - 2p) \cdot (1 + 4\delta).
\end{aligned}$$

Since $\rho(\mathbf{x})$ is monotonically increasing with respect to $\delta$ and $\delta \in [0, \alpha]$, it follows that $\rho(\mathbf{x})$ is at most $2p \cdot \frac{1+2\alpha}{1-2\alpha} + (1 - 2p) \cdot (1 + 4\alpha)$ in which $\delta = \alpha$, i.e., when $\mathrm{mid}(\mathbf{x}) = \frac{1}{2} - \alpha$, the approximation ratio is maximized, and the worst case falls into the instance $(0, 1 - 2\alpha)$ where the inequalities become equality.

**Case (2).** $\delta \in (\alpha, \frac{1}{2}]$, i.e., $0 \leq \mathrm{mid}(\mathbf{x}) < \frac{1}{2} - \alpha$, we first introduce the following claim.

**Claim B.1.** When $\mathrm{mid}(\mathbf{x}) \in [0, \frac{1}{2} - \alpha)$, for any instance $\mathbf{x} = (x_1, x_2)$, the approximation ratio of $(\alpha, p)$-LRM mechanism under $\mathbf{x}$ is upper-bounded by the approximation ratio under $\mathbf{x}' = (0, x_2)$.

*Proof.* The proof starts by observing that for any instance $\mathbf{x} = (x_1, x_2)$, it holds that $y - x_1 \geq |y - x_2|$ for any $y \in \{\frac{1}{2} - \alpha, \frac{1}{2}, \frac{1}{2} + \alpha\}$. That is, agent 1 always obtains a smaller utility than agent 2. We prove this by considering the location of $x_2$ and each potential facility location $y$. If $x_2 \leq y$, the expression trivially holds as $x_2 \geq x_1$. Conversely, we have $x_2 - y \leq y - x_1$ as $\mathrm{mid}(\mathbf{x}) = \frac{x_1 + x_2}{2} \leq \frac{1}{2} - \alpha$ and $y \geq \frac{1}{2} - \alpha$.

Let $u_1(y)$ and $u_2(y)$ be the utilities of agent 1 and 2 under a potential facility location $y \in \{\frac{1}{2} - \alpha, \frac{1}{2}, \frac{1}{2} + \alpha\}$. As we know that $u_1(y) \leq u_2(y)$ for each $y \in \{\frac{1}{2} - \alpha, \frac{1}{2}, \frac{1}{2} + \alpha\}$, the approximation ratio of $(\alpha, p)$-LRM mechanism is upper-bounded by

$$\begin{aligned}
\rho(\mathbf{x}) &= p \cdot \frac{u_2(\frac{1}{2} - \alpha)}{u_1(\frac{1}{2} - \alpha)} + (1 - 2p) \cdot \frac{u_2(\frac{1}{2})}{u_1(\frac{1}{2})} + p \cdot \frac{u_2(\frac{1}{2} + \alpha)}{u_1(\frac{1}{2} + \alpha)} \\
&= p \cdot \frac{u_2(\frac{1}{2} - \alpha)}{1 - ((\frac{1}{2} - \alpha) - x_1)} + (1 - 2p) \cdot \frac{u_2(\frac{1}{2})}{1 - (\frac{1}{2} - x_1)} + p \cdot \frac{u_2(\frac{1}{2} + \alpha)}{1 - ((\frac{1}{2} + \alpha) - x_1)} \\
&\leq p \cdot \frac{u_2(\frac{1}{2} - \alpha)}{1 - ((\frac{1}{2} - \alpha) - x_1')} + (1 - 2p) \cdot \frac{u_2(\frac{1}{2})}{1 - (\frac{1}{2} - x_1')} + p \cdot \frac{u_2(\frac{1}{2} + \alpha)}{1 - ((\frac{1}{2} + \alpha) - x_1')} = \rho(\mathbf{x}').
\end{aligned}$$

$\square$

With the claim in hand, we now turn our attention into the instances $\mathbf{x} = (0, x_2)$ when $\mathrm{mid}(\mathbf{x}) \in [0, \frac{1}{2} - \alpha]$. We divide the proof into subcases depending on the location of $x_2$.

**Sub-Case (a).** $x_2 \in [0, \frac{1}{2} - \alpha]$. The approximation ratio is represented as

$$\rho(\mathbf{x}) = p \cdot \frac{1 - (\frac{1}{2} - \alpha - x_2)}{1 - (\frac{1}{2} - \alpha)} + (1 - 2p) \cdot \frac{1 - (\frac{1}{2} - x_2)}{1 - \frac{1}{2}} + p \cdot \frac{1 - (\frac{1}{2} + \alpha - x_2)}{1 - (\frac{1}{2} + \alpha)}$$

$$= p \cdot \frac{1 + 2\alpha + 2x_2}{1 + 2\alpha} + (1 - 2p) \cdot (1 + 2x_2) + p \cdot \frac{1 - 2\alpha + 2x_2}{1 - 2\alpha}.$$

Here, $\rho(\mathbf{x})$ has a derivative of

$$\frac{\mathrm{d}\rho(\mathbf{x})}{\mathrm{d}x_2} = \frac{2p}{1 + 2\alpha} + 2(1 - 2p) + \frac{2p}{1 - 2\alpha} \geq 0,$$

as $p \in [0, \frac{1}{2}]$ and $\alpha \in [0, \frac{1}{6}]$. This implies that $\rho(\mathbf{x})$ is monotonically increasing with respect to $x_2$. Since $x_2 \in [0, \frac{1}{2} - \alpha]$, we have that the approximation ratio of any instance $\mathbf{x}$ where $x_2 \in [0, \frac{1}{2} - \alpha]$ is maximized under the instance $\mathbf{x} = (x_1 = 0, x_2 = \frac{1}{2} - \alpha)$.

**Sub-Case (b).** $x_2 \in (\frac{1}{2} - \alpha, \frac{1}{2}]$. We slightly modify the approximation ratio expression from sub-case (a) and get

$$\rho(\mathbf{x}) = p \cdot \frac{1 - (x_2 - (\frac{1}{2} - \alpha))}{1 - (\frac{1}{2} - \alpha)} + (1 - 2p) \cdot \frac{1 - (\frac{1}{2} - x_2)}{1 - \frac{1}{2}} + p \cdot \frac{1 - (\frac{1}{2} + \alpha - x_2)}{1 - (\frac{1}{2} + \alpha)}$$

$$= p \cdot \frac{3 - 2x_2 - 2\alpha}{1 + 2\alpha} + (1 - 2p) \cdot (1 + 2x_2) + p \cdot \frac{1 - 2\alpha + 2x_2}{1 - 2\alpha}.$$

Similarly, we compute the derivative of $\rho(\mathbf{x})$ with respect to $x_2$

$$\frac{\mathrm{d}\rho(\mathbf{x})}{\mathrm{d}x_2} = -\frac{2p}{1 + 2\alpha} + 2(1 - 2p) + \frac{2p}{1 + 2\alpha} = \frac{8p\alpha}{1 - 4\alpha^2} + 2(1 - 2p) \geq 0.$$

$$(\because p \in [0, \tfrac{1}{2}], \alpha \in [0, \tfrac{1}{6}])$$

It follows that the instance with the maximum approximation ratio has $x_2 = \frac{1}{2}$, i.e., $\mathbf{x} = (x_1 = 0, x_2 = \frac{1}{2})$.

**Sub-Case (c).** $x_2 \in (\frac{1}{2}, \frac{1}{2} + \alpha]$. In this case, we have

$$\rho(\mathbf{x}) = p \cdot \frac{1 - (x_2 - (\frac{1}{2} - \alpha))}{1 - (\frac{1}{2} - \alpha)} + (1 - 2p) \cdot \frac{1 - (x_2 - \frac{1}{2})}{1 - \frac{1}{2}} + p \cdot \frac{1 - (\frac{1}{2} + \alpha - x_2)}{1 - (\frac{1}{2} + \alpha)}$$

$$= p \cdot \frac{3 - 2x_2 - 2\alpha}{1 + 2\alpha} + (1 - 2p) \cdot (3 - 2x_2) + p \cdot \frac{1 - 2\alpha + 2x_2}{1 - 2\alpha}.$$

The derivative is written as

$$\frac{\mathrm{d}\rho(\mathbf{x})}{\mathrm{d}x_2} = -\frac{2p}{1 + 2\alpha} - 2(1 - 2p) + \frac{2p}{1 - 2\alpha} = \frac{8p\alpha}{1 - 4\alpha^2} + 4p - 2.$$

Notably, when given $\alpha$ and $p$, the derivative is a constant, implying that the approximation ratio is either monotonically increasing or decreasing with respect to $x_2$. That is, for any instance in this sub-case, the approximation ratio is either upper-bounded by that under $\mathbf{x} = (x_1 = 0, x_2 = \frac{1}{2})$ or $\mathbf{x} = (x_1 = 0, x_2 = \frac{1}{2} + \alpha)$.

**Sub-Case (d).** $x_2 \in (\frac{1}{2} + \alpha, 1 - 2\alpha]$. The approximation ratio is computed as

$$\rho(\mathbf{x}) = p \cdot \frac{1 - (x_2 - (\frac{1}{2} - \alpha))}{1 - (\frac{1}{2} - \alpha)} + (1 - 2p) \cdot \frac{1 - (x_2 - \frac{1}{2})}{1 - \frac{1}{2}} + p \cdot \frac{1 - (x_2 - (\frac{1}{2} + \alpha))}{1 - (\frac{1}{2} + \alpha)}$$

$$= p \cdot \frac{3 - 2x_2 - 2\alpha}{1 + 2\alpha} + (1 - 2p) \cdot (3 - 2x_2) + p \cdot \frac{3 + 2\alpha - 2x_2}{1 - 2\alpha}.$$

The derivative of $\rho(\mathbf{x})$ w.r.t. $x_2$ is

$$\frac{\mathrm{d}\rho(\mathbf{x})}{\mathrm{d}x_2} = -\frac{2p}{1 + 2\alpha} - 2(1 - 2p) - \frac{2p}{1 - 2\alpha} = -\frac{4p}{1 - 4\alpha^2} - 2(1 - 2p) \leq 0.$$

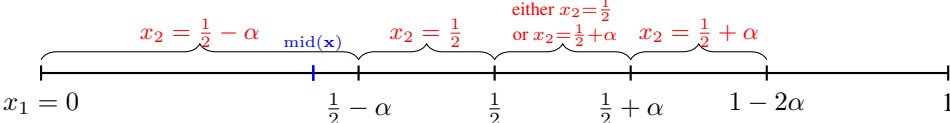

$$x_1 = 0 \qquad \tfrac{1}{2} - \alpha \qquad \tfrac{1}{2} \qquad \tfrac{1}{2} + \alpha \qquad 1 - 2\alpha \qquad 1$$

Figure 4: Summary of Analysis when $\mathrm{mid}(\mathbf{x}) \in [0, \tfrac{1}{2} - \alpha]$

Since $\rho(\mathbf{x})$ is monotonically non-increasing with respect to $x_2$ and $x_2 \in (\tfrac{1}{2} + \alpha, 1 - 2\alpha]$, it follows that $\rho(\mathbf{x})$ is upper-bounded by the approximation ratio under the instance $\mathbf{x} = (x_1 = 0, x_2 = \tfrac{1}{2} + \alpha)$.

Figure 4 depicts the worst instances under $4$ sub-cases in Case (2).

We next observe that the approximation ratio of instance $\mathbf{x} = (x_1 = 0, x_2 = \tfrac{1}{2} - \alpha)$ is no worse than that of instance $\mathbf{x} = (x_1 = 0, x_2 = \tfrac{1}{2})$, and the approximation ratio of instance $\mathbf{x} = (x_1 = 0, x_2 = 1 - 2\alpha)$ under Case (1) is no worse than that of instance $\mathbf{x} = (x_1 = 0, x_2 = \tfrac{1}{2} + \alpha)$.

By combining all aforementioned subcase discussions, we conclude that when $\alpha \leq \tfrac{1}{6}$, computing the general optimal $(\alpha, p)$-LRM mechanism boils down to finding the optimal $(\alpha, p)$ which optimizes the approximation ratio of envy ratio objective under instance $\mathbf{x} = (x_1 = 0, x_2 = \tfrac{1}{2})$ and $\mathbf{x}' = (x_1' = 0, x_2' = \tfrac{1}{2} + \alpha)$.

We next consider the remaining main case where $\alpha \in [\tfrac{1}{6}, \tfrac{1}{4}]$. We consider three subcases depending on the position of $\mathrm{mid}(\mathbf{x})$. Since we only consider $\mathrm{mid}(\mathbf{x}) \in [0, \tfrac{1}{2}]$, the approximation ratio $\rho(\mathbf{x})$ is viewed as a function of $u$ (recall $u$ is the optimal utility under $\mathrm{mid}(\mathbf{x})$), which is monotonically decreasing with respect to $u$. Note that $u \geq \tfrac{1}{2} + \delta$, which implies that we only need to consider the cases where $x_1 = 0$. By fixing $x_1 = 0$, we mainly consider the location of $x_2$ as follows.

**Case (1).** $x_2 \in [0, \tfrac{1}{2} - \alpha]$. The approximation ratio is represented as

$$\rho(\mathbf{x}) = p \cdot \frac{1 - (\tfrac{1}{2} - \alpha - x_2)}{1 - (\tfrac{1}{2} - \alpha)} + (1 - 2p) \cdot \frac{1 - (\tfrac{1}{2} - x_2)}{1 - \tfrac{1}{2}} + p \cdot \frac{1 - (\tfrac{1}{2} + \alpha - x_2)}{1 - (\tfrac{1}{2} + \alpha)}$$

$$= p \cdot \frac{1 + 2\alpha + 2x_2}{1 + 2\alpha} + (1 - 2p) \cdot (1 + 2x_2) + p \cdot \frac{1 - 2\alpha + 2x_2}{1 - 2\alpha},$$

which is monotonically increasing with respect to $x_2$, implying the worst instance in this case is $\mathbf{x} = (x_1 = 0, x_2 = \tfrac{1}{2} - \alpha)$.

**Case (2).** $x_2 \in (\tfrac{1}{2} - \alpha, \tfrac{1}{2}]$. The approximation ratio is represented as

$$\rho(\mathbf{x}) = p \cdot \frac{1 - (x_2 - (\tfrac{1}{2} - \alpha))}{1 - (\tfrac{1}{2} - \alpha)} + (1 - 2p) \cdot \frac{1 - (\tfrac{1}{2} - x_2)}{1 - \tfrac{1}{2}} + p \cdot \frac{1 - (\tfrac{1}{2} + \alpha - x_2)}{1 - (\tfrac{1}{2} + \alpha)}$$

$$= p \cdot \frac{3 - 2x_2 - 2\alpha}{1 + 2\alpha} + (1 - 2p) \cdot (1 + 2x_2) + p \cdot \frac{1 - 2\alpha + 2x_2}{1 - 2\alpha}.$$

Consequently, the derivative of $\rho(\mathbf{x})$ over $x_2$ is

$$\frac{\mathrm{d}\rho(\mathbf{x})}{\mathrm{d}x_2} = -\frac{2p}{1 + 2\alpha} + 2(1 - 2p) + \frac{2p}{1 - 2\alpha} = \frac{8p\alpha}{1 - 4\alpha^2} + 2(1 - 2p) \geq 0.$$

Therefore, the approximation ratio is upper-bounded by the instance where $x_1 = 0, x_2 = \tfrac{1}{2}$.

**Case (3).** $x_2 \in (\tfrac{1}{2}, 1 - 2\alpha]$. The approximation ratio is represented as

$$\rho(\mathbf{x}) = p \cdot \frac{1 - (x_2 - (\tfrac{1}{2} - \alpha))}{1 - (\tfrac{1}{2} - \alpha)} + (1 - 2p) \cdot \frac{1 - (x_2 - \tfrac{1}{2})}{1 - \tfrac{1}{2}} + p \cdot \frac{1 - (\tfrac{1}{2} + \alpha - x_2)}{1 - (\tfrac{1}{2} + \alpha)}$$

$$= p \cdot \frac{3 - 2x_2 - 2\alpha}{1 + 2\alpha} + (1 - 2p) \cdot (3 - 2x_2) + p \cdot \frac{1 - 2\alpha + 2x_2}{1 - 2\alpha}.$$

We compute the derivative of $\rho(\mathbf{x})$ with respect to $x_2$

$$\frac{\mathrm{d}\rho(\mathbf{x})}{\mathrm{d}x_2} = -\frac{2p}{1 + 2\alpha} - 2(1 - 2p) + \frac{2p}{1 - 2\alpha}.$$

**Case (4).** $x_2 \in (1 - 2\alpha, \frac{1}{2} + \alpha]$. The approximation ratio is represented as

$$\rho(\mathbf{x}) = p \cdot \frac{1 - (\frac{1}{2} - \alpha)}{1 - (x_2 - (\frac{1}{2} - \alpha))} + (1 - 2p) \cdot \frac{1 - (x_2 - \frac{1}{2})}{1 - \frac{1}{2}} + p \cdot \frac{1 - (\frac{1}{2} + \alpha - x_2)}{1 - (\frac{1}{2} + \alpha)}$$

$$= p \cdot \frac{1 + 2\alpha}{3 - 2x_2 - 2\alpha} + (1 - 2p) \cdot (3 - 2x_2) + p \cdot \frac{1 - 2\alpha + 2x_2}{1 - 2\alpha}.$$

Similarly, the derivative of $\rho(\mathbf{x})$ over $x_2$ is written as

$$\frac{\mathrm{d}\rho(\mathbf{x})}{\mathrm{d}x_2} = \frac{2p(1 + 2\alpha)}{(3 - 2x_2 - 2\alpha)^2} - 2(1 - 2p) + \frac{2p}{1 - 2\alpha}.$$

Notably, the derivative in Case (4) is no less than that of Case (3). It follows that if $\rho(\mathbf{x})$ is monotonically increasing w.r.t. $x_2$ in Case (3), then $\rho(\mathbf{x})$ is monotonically increasing w.r.t. $x_2$ in Case (4). Conversely, if $\rho(\mathbf{x})$ is monotonically decreasing w.r.t. $x_2$ in Case (3), $\rho(\mathbf{x})$ could be either monotonically increasing or decreasing. This implies that the instance with worst approximation ratio is either $x_2 = \frac{1}{2}$ or $x_2 = \frac{1}{2} + \alpha$.

**Case (5).** $x_2 \in (\frac{1}{2} + \alpha, 1]$. Recall the definition of $\delta = \frac{1}{2} - \mathrm{mid}(\mathbf{x})$, we write the approximation ratio $\rho(\mathbf{x})$ as a function of $\delta$

$$\rho(\mathbf{x}) \leq p \cdot \frac{\frac{1}{2} + \alpha}{\frac{1}{2} + 2\delta - \alpha} + (1 - 2p) \cdot \frac{\frac{1}{2} + 2\delta}{\frac{1}{2}} + p \cdot \frac{\frac{1}{2} + 2\delta + \alpha}{\frac{1}{2} - \alpha}$$

$$= p \cdot \frac{1 + 2\alpha}{1 + 4\delta - 2\alpha} + (1 - 2p) \cdot (1 + 4\delta) + p \cdot \frac{1 + 4\delta + 2\alpha}{1 - 2\alpha}.$$

Since $x_2 \in (\frac{1}{2} + \alpha, 1]$, we have $\delta \in [0, \frac{1}{4} - \frac{\alpha}{2}]$. Likewise, we use the same technique in Case (1) when considering $\alpha \in [0, \frac{1}{6}]$. From the convexity of the term $\frac{1+2\alpha}{1+4\delta-2\alpha}$, we get

$$\rho(\mathbf{x}) \leq p \cdot \left(-\frac{2 + 4\alpha}{(1 - 2\alpha)^2}\delta + \frac{1 + 2\alpha}{1 - 2\alpha}\right) + (1 - 2p) \cdot (1 + 4\delta) + p \cdot \frac{1 + 4\delta + 2\alpha}{1 - 2\alpha}.$$

Consider the derivative of the RHS.

$$\frac{\mathrm{d}\rho(\mathbf{x})}{\mathrm{d}\delta} = -\frac{(2 + 4\alpha)p}{(1 - 2\alpha)^2} + 4(1 - 2p) + \frac{4p}{1 - 2\alpha}$$

$$\geq -\frac{2 + \frac{2}{3}}{\frac{4}{9}}p + 4(1 - 2p) + \frac{4p}{1 - \frac{1}{3}} \qquad (\because \text{ monotonically increasing w.r.t. } \alpha \in [\frac{1}{6}, \frac{1}{4}])$$

$$= 4 - 8p \geq 0.$$

This implies that the RHS is monotonically increasing with respect to $\delta$ when $\delta \in [0, \frac{1}{4} - \frac{\alpha}{2}]$. Hence, $\rho(\mathbf{x})$ is upper-bounded by the instance when $\delta = \frac{1}{4} - \frac{\alpha}{2}$, i.e., $\mathbf{x} = (x_1 = 0, x_2 = \frac{1}{2} + \alpha)$.

By combining the two main cases, i.e., $\alpha \in [0, \frac{1}{6})$ and $\alpha \in [\frac{1}{6}, \frac{1}{4}]$, we derive that $\mathbf{x} = (x_1 = 0, x_2 = \frac{1}{2})$ and $\mathbf{x}' = (x_1' = 0, x_2' = \frac{1}{2} + \alpha)$ are the two instances with the worst approximation ratio. Formally, when $\alpha \leq \frac{1}{4}$, the $(\alpha^*, p^*)$-LRM constant Mechanism optimizes the approximation ratio of envy ratio objective where $(\alpha^*, p^*) = \arg\min_{(\alpha, p)}\{\max\{\rho(\mathbf{x}), \rho(\mathbf{x}')\}\}$. This completes the proof. $\square$

### B.3 Proof of Theorem 4.3

*Proof.* Anonymity and strategyproofness are immediate. From Lemma 4.1 and Lemma 4.2, it suffices to only consider the two special instances $\mathbf{x} = (x_1 = 0, x_2 = \frac{1}{2})$, and $\mathbf{x}' = (x_1' = 0, x_2' = \frac{1}{2} + \alpha)$.

Consider any arbitrary $(\alpha, p)$-LRM mechanism where $\alpha \in [0, \frac{1}{4})$. We express the approximation ratio for instances $\mathbf{x}$ and $\mathbf{x}'$ as a function of $\alpha$ and $p$ as follows. The approximation ratio under $\mathbf{x}$ can be expressed as

$$\rho(\mathbf{x}) = p \cdot \frac{1 - \alpha}{1 - (\frac{1}{2} - \alpha)} + (1 - 2p) \cdot \frac{1 - 0}{1 - \frac{1}{2}} + p \cdot \frac{1 - \alpha}{1 - (\frac{1}{2} + \alpha)}.$$

The expression of the approximation ratio under $\mathbf{x}'$ slightly varies depending on the range of $\alpha$. In particular, when $\alpha \in [0, \frac{1}{6}]$, it can be represented as

$$\rho(\mathbf{x}') = p \cdot \frac{1 - 2\alpha}{\frac{1}{2} + \alpha} + (1 - 2p) \cdot \frac{1 - \alpha}{\frac{1}{2}} + p \cdot \frac{1}{\frac{1}{2} - \alpha}$$

$$= p \cdot \frac{2 - 4\alpha}{1 + 2\alpha} + (1 - 2p)(2 - 2\alpha) + p \cdot \frac{2}{1 - 2\alpha}.$$

When $\alpha \in (\frac{1}{6}, \frac{1}{4}]$, for instance $\mathbf{x}'$, when placing the facility at $\frac{1}{2} - \alpha$, $x_1$ is the agent who has higher utility. Henceforth, the approximation ratio is written as

$$\rho(\mathbf{x}') = p \cdot \frac{\frac{1}{2} + \alpha}{1 - 2\alpha} + (1 - 2p) \cdot \frac{1 - \alpha}{\frac{1}{2}} + p \cdot \frac{1}{\frac{1}{2} - \alpha}$$

$$= p \cdot \frac{1 + 2\alpha}{2 - 4\alpha} + (1 - 2p)(2 - 2\alpha) + p \cdot \frac{2}{1 - 2\alpha}.$$

Since the approximation ratio under instance $\mathbf{x}'$ varies with respect to the range of $\alpha$. We compute the optimal parameters of $\alpha$ and $p$ by considering $\alpha \in [0, \frac{1}{6}]$ and $\alpha \in (\frac{1}{6}, \frac{1}{4}]$, respectively.

**Case (1).** $\alpha \in [0, \frac{1}{6}]$. We show that $\min_{\alpha \in [0, \frac{1}{6}), p \in [0, \frac{1}{2}]} \max\{\rho(\mathbf{x}), \rho(\mathbf{x}')\}$ takes an optimal value of approximately $1 + \frac{2}{\sqrt{5}} \approx 1.8944$ when $\alpha = \frac{\sqrt{5}}{2} - 1$ and $p = \frac{2}{5}$.

We first consider the values of $\alpha$ and $p$ which satisfy $\rho(\mathbf{x}) = \rho(\mathbf{x}')$.

We have

$$2(1 - 2p) + p \cdot \frac{4 - 4\alpha}{1 - 4\alpha^2} = p \cdot \frac{2 - 4\alpha}{1 + 2\alpha} + (1 - 2p)(2 - 2\alpha) + p \cdot \frac{2}{1 - 2\alpha}.$$

Dividing both sides by 2 simplifies the expression to

$$1 - 2p + p \cdot \frac{2 - 2\alpha}{1 - 4\alpha^2} = p \cdot \frac{1 - 2\alpha}{1 + 2\alpha} + (1 - 2p)(1 - \alpha) + p \cdot \frac{1}{1 - 2\alpha},$$

$$\iff \alpha - 2\alpha p = \frac{p(1 - 2\alpha)^2}{1 - 4\alpha^2} + \frac{p(1 + 2\alpha)}{1 - 4\alpha^2} - \frac{p(2 - 2\alpha)}{1 - 4\alpha^2}$$

$$\iff (\alpha - 2\alpha p)(1 - 4\alpha^2) = p(1 - 4\alpha + 4\alpha^2 + 1 + 2\alpha - 2 + 2\alpha)$$

$$\iff \alpha - 4\alpha^3 - 2\alpha p + 8\alpha^3 p = 4\alpha^2 p$$

$$\iff p(8\alpha^2 - 4\alpha^2 - 2\alpha) = 4\alpha^3 - \alpha$$

$$\iff p = \frac{4\alpha^2 - 1}{2(4\alpha^2 - 2\alpha - 1)}.$$

Hence, we know that $\rho(\mathbf{x}) = \rho(\mathbf{x}')$ when $p = \frac{4\alpha^2 - 1}{2(4\alpha^2 - 2\alpha - 1)}$. Note that this solution also requires $4\alpha^2 - 1 \neq 0$ and $4\alpha^2 - 2\alpha - 1 \neq 0$, which are achieved under $\alpha \in [0, \frac{1}{6})$. Substituting $p = \frac{4\alpha^2 - 1}{2(4\alpha^2 - 2\alpha - 1)}$ into $2(1 - 2p) + p \cdot \frac{4 - 4\alpha}{1 - 4\alpha^2}$ gives us

$$2\left(1 - \frac{4\alpha^2 - 1}{4\alpha^2 - 2\alpha - 1}\right) + \frac{4\alpha^2 - 1}{2(4\alpha^2 - 2\alpha - 1)} \cdot \frac{4 - 4\alpha}{1 - 4\alpha^2} = \frac{8\alpha^2 - 4\alpha - 2 - 8\alpha^2 + 2}{4\alpha^2 - 2\alpha - 1} + \frac{2\alpha - 2}{4\alpha^2 - 2\alpha - 1}$$

$$= \frac{2\alpha + 2}{-4\alpha^2 + 2\alpha + 1},$$

which has a derivative of

$$\frac{\mathrm{d}}{\mathrm{d}\alpha}\left(\frac{2\alpha + 2}{-4\alpha^2 + 2\alpha + 1}\right) = \frac{8\alpha^2 + 16\alpha - 2}{(-4\alpha^2 + 2\alpha + 1)^2}.$$

This derivative is equal to 0 when $\alpha = -1 - \frac{\sqrt{5}}{2}$ or when $\alpha = \frac{\sqrt{5}}{2} - 1 \approx 0.118$. We ignore the former value as $\alpha \geq 0$. Substituting $\alpha = \frac{\sqrt{5}}{2} - 1$ into $p = \frac{4\alpha^2 - 1}{2(4\alpha^2 - 2\alpha - 1)}$ gives $p = \frac{2}{5}$.

By substitution, we have that $\rho(\mathbf{x}) = \rho(\mathbf{x}') = 1 + \frac{2}{\sqrt{5}} \approx 1.8944$ when $\alpha = \frac{\sqrt{5}}{2} - 1$ and $p = \frac{2}{5}$. From Lemmas B.2 and B.3, it follows that $\min_{\alpha \in [0, \frac{1}{6}), p \in [0, \frac{1}{2}]} \max\{\rho(\mathbf{x}), \rho(\mathbf{x}')\} = 1 + \frac{2}{\sqrt{5}}$, proving the optimality of the mechanism.

**Case (1).** $\alpha \in [\frac{1}{6}, \frac{1}{4}]$. With a similar method of analysis in (1), we can prove that $\min_{\alpha \in [\frac{1}{6}, \frac{1}{4}], p \in [0, \frac{1}{2}]} \max\{\rho(\mathbf{x}), \rho(\mathbf{x}')\}$ takes an optimal value of approximately $\frac{21}{11} \approx 1.8944$ when $\alpha = \frac{1}{6}$ and $p = \frac{4}{11}$. Similarly, we first consider the value of $\alpha$ and $p$ which satisfy $\rho(\mathbf{x}) = \rho(\mathbf{x}')$. That is,

$$2(1 - 2p) + p \cdot \frac{4 - 4\alpha}{1 - 4\alpha^2} = p \cdot \frac{1 + 2\alpha}{2 - 4\alpha} + (1 - 2p)(2 - 2\alpha) + p \cdot \frac{2}{1 - 2\alpha}.$$

It follows that

$$p = \frac{16\alpha^3 - 4\alpha}{32\alpha^3 - 4\alpha^2 - 28\alpha + 3}.$$

Henceforth, $\rho(\mathbf{x}) = \rho(\mathbf{x}')$ when $p = \frac{16\alpha^3 - 4\alpha}{32\alpha^3 - 4\alpha^2 - 28\alpha + 3}$. By substituting the $p$ back into $\rho(\mathbf{x})$, we have

$$\rho(\mathbf{x}) = \rho(\mathbf{x}') = \frac{8\alpha^2 - 56\alpha + 6}{32\alpha^3 - 4\alpha^2 - 28\alpha + 3},$$

which is monotonically increasing with respect to $\alpha \in [\frac{1}{6}, \frac{1}{4}]$. Hence, by leveraging the very similar technique as in (1), we obtain that when $\alpha = \frac{1}{6}$ and $p = \frac{4}{11}$, $\min_{\alpha \in [\frac{1}{6}, \frac{1}{4}]} \max\{\rho(\mathbf{x}), \rho(\mathbf{x}')\} = \frac{21}{11} \approx 1.909$.

By combining these two case analysis, we conclude that the $(\frac{\sqrt{5}}{2} - 1, \frac{2}{5})$-LRM constant mechanism optimizes the approximation ratio at $1 + \frac{2}{\sqrt{5}} \approx 1.8944$. $\qquad\square$

**Lemma B.2.** *If $\rho(\mathbf{x}) < 1 + \frac{2}{\sqrt{5}}$, then $\rho(\mathbf{x}') > 1 + \frac{2}{\sqrt{5}}$.*

*Proof.* We have

$$\rho(\mathbf{x}) < 1 + \frac{2}{\sqrt{5}}$$

$$\iff 2 - 4p + p \cdot \frac{4 - 4\alpha}{1 - 4\alpha^2} < 1 + \frac{2}{\sqrt{5}}$$

$$\iff 4p\left(1 - \frac{1 - \alpha}{1 - 4\alpha^2}\right) > 1 - \frac{2}{\sqrt{5}}$$

$$\iff p > \frac{(1 - \frac{2}{\sqrt{5}})(1 - 4\alpha^2)}{4\alpha(1 - 4\alpha)}.$$

Note that the approximation ratio under $\mathbf{x}'$ can be rewritten as

$$\rho(\mathbf{x}') = p \cdot \frac{2 - 4\alpha}{1 + 2\alpha} + (1 - 2p)(2 - 2\alpha) + p \cdot \frac{2}{1 - 2\alpha}$$

$$= p\left(\frac{2 - 4\alpha}{1 + 2\alpha} + 4\alpha - 4 + \frac{2}{1 - 2\alpha}\right) + 2 - 2\alpha$$

$$= p\left(\frac{8\alpha^2(3 - 2\alpha)}{(1 - 2\alpha)(2\alpha + 1)}\right) + 2 - 2\alpha.$$

Since $p \cdot \frac{2 - 4\alpha}{1 + 2\alpha} + (1 - 2p)(2 - 2\alpha) + p \cdot \frac{2}{1 - 2\alpha} \geq 0$ for all $0 \leq \alpha \leq \frac{1}{6}$, we can substitute $p > \frac{(1 - \frac{2}{\sqrt{5}})(1 - 4\alpha^2)}{4\alpha(1 - 4\alpha)}$ to obtain

$$\rho(\mathbf{x}') > \left(\frac{(1 - \frac{2}{\sqrt{5}})(1 - 4\alpha^2)}{4\alpha(1 - 4\alpha)}\right)\left(\frac{8\alpha^2(3 - 2\alpha)}{(1 - 2\alpha)(2\alpha + 1)}\right) + 2 - 2\alpha$$

$$= \frac{(1 - \frac{2}{\sqrt{5}}) \cdot 2\alpha(3 - 2\alpha)}{1 - 4\alpha} + 2 - 2\alpha.$$

The derivative of this expression is

$$\frac{d}{d\alpha}\left(\frac{(1-\frac{2}{\sqrt{5}})\cdot 2\alpha(3-2\alpha)}{1-4\alpha}+2-2\alpha\right)=-\frac{4(4(5+2\sqrt{5})\alpha^2-2(5+2\sqrt{5})\alpha+3\sqrt{5}-5}{5(1-4\alpha)^2},$$

which is equal to 0 when $\alpha=\frac{\sqrt{5}}{2}-1$ or $\alpha=\frac{3-\sqrt{5}}{2}$. Thus, we see that when $\alpha\in[0,\frac{1}{6}]$, the expression takes a minimum of $1+\frac{2}{\sqrt{5}}$ when $\alpha=\frac{\sqrt{5}}{2}-1$. Therefore, $\rho(\mathbf{x}')>1+\frac{2}{\sqrt{5}}$ when $\rho(\mathbf{x})<1+\frac{2}{\sqrt{5}}$. $\square$

**Lemma B.3.** *If $\rho(\mathbf{x}')<1+\frac{2}{\sqrt{5}}$, then $\rho(\mathbf{x})>1+\frac{2}{\sqrt{5}}$.*

*Proof.* We have

$$\rho(\mathbf{x}')<1+\frac{2}{\sqrt{5}}$$

$$\iff p\left(\frac{8\alpha^2(3-2\alpha)}{(1-2\alpha)(2\alpha+1)}\right)<2\alpha-1+\frac{2}{\sqrt{5}}$$

$$\iff p<\frac{(2\alpha-1+\frac{2}{\sqrt{5}})(1-4\alpha^2)}{8\alpha^2(3-2\alpha)}.$$

Note that when $\alpha<\frac{1}{2}-\frac{1}{\sqrt{5}}$, the RHS becomes negative and consequently, the inequality cannot be satisfied. We therefore restrict our attention to $\alpha\in[\frac{1}{2}-\frac{1}{\sqrt{5}},\frac{1}{6}]$.

Note that $\rho(\mathbf{x})=2+4p\left(\frac{4\alpha^2-\alpha}{1-4\alpha^2}\right)$, and that $\frac{4\alpha^2-\alpha}{1-4\alpha^2}<0$ when $\alpha\in[\frac{1}{2}-\frac{1}{\sqrt{5}},\frac{1}{6}]$. Therefore, by substituting $p<\frac{(2\alpha-1+\frac{2}{\sqrt{5}})(1-4\alpha^2)}{8\alpha^2(3-2\alpha)}$, we have that when $\rho(\mathbf{x}')<1+\frac{2}{\sqrt{5}}$,

$$\rho(\mathbf{x})>2+\frac{(2\alpha-1+\frac{2}{\sqrt{5}})(4\alpha-1)}{2\alpha(3-2\alpha)}.$$

The derivative of the RHS is

$$\frac{d}{da}\left(2+\frac{(2\alpha-1+\frac{2}{\sqrt{5}})(4\alpha-1)}{2\alpha(3-2\alpha)}\right)=\frac{4(15+4\sqrt{5})\alpha^2+(20-8\sqrt{5})\alpha+6\sqrt{5}-15}{10(3-2\alpha)^2\alpha^2},$$

which is equal to 0 when $\alpha=\frac{\sqrt{5}}{2}-1$ or when $\alpha=\frac{6}{29}-\frac{9\sqrt{5}}{58}$. We therefore see that when $\alpha\in[0,\frac{1}{6}]$, the expression takes a minimum of $1+\frac{2}{\sqrt{5}}$ when $\alpha=\frac{\sqrt{5}}{2}-1$, proving that $\rho(\mathbf{x})>1+\frac{2}{\sqrt{5}}$ when $\rho(\mathbf{x}')<1+\frac{2}{\sqrt{5}}$. $\square$

### B.4 Proof of Theorem 4.4

*Proof.* To show the lower bound, we first consider the location profile with two agents $\mathbf{x}=(x_1=0.29,x_2=0.71)$. Note that for any randomized mechanism $f$, we have either $\mathbb{E}_{y\in f(\mathbf{x})}[|y-0.29|]\geq 0.21$ or $\mathbb{E}_{y\in f(\mathbf{x})}[|y-0.71|]\geq 0.21$.

We first consider the former case, where $\mathbb{E}_{y\in f(\mathbf{x})}[|y-0.29|]\geq 0.21$. Let $\mathbf{x}'=(0,0.71)$. Then we must have

$$\mathbb{E}_{y\in f(\mathbf{x}')}[|y-0.29|]\geq\mathbb{E}_{y\in f(\mathbf{x})}[|y-0.29|]\geq 0.21,$$

otherwise agent 1 at $x_1=0.29$ has an incentive to change her reported location to $x_i'=0$ for a better outcome, violating strategyproofness. Denote $\delta:=\frac{617}{4300}\approx 0.14$, and

$$p_1:=\Pr\{f(\mathbf{x}')\in[0.29-\delta,0.29+\delta]\},$$
$$p_2:=\Pr\{f(\mathbf{x}')\in[0,0.29-\delta)\cup(0.29+\delta,0.58]\},$$
$$p_3:=\Pr\{f(\mathbf{x}')\in(0.58,1]\}.$$

Then, we have

$$\mathbb{E}_{y \in f(\mathbf{x}')}[|y - 0.29|] \leq \delta p_1 + 0.29 p_2 + 0.71 p_3$$
$$= \delta + (0.29 - \delta)p_2 + (0.71 - \delta)p_3.$$

Substituting $\mathbb{E}_{y \in f(\mathbf{x}')}[|y - 0.29|] \geq 0.21$ gives us

$$\delta + (0.29 - \delta)p_2 + (0.71 - \delta)p_3 \geq 0.21,$$

which we can rearrange to form the inequality

$$p_3 \geq \frac{0.21 - \delta - (0.29 - \delta)p_2}{0.71 - \delta}.$$

Finally, we have

$$\mathrm{ER}(f(\mathbf{x}'), \mathbf{x}')$$
$$\geq p_1 + \frac{1 - (0.71 - (0.29 + \delta))}{1 - (0.29 + \delta - 0)}p_2 + \frac{1 - (0.71 - 0.58)}{1 - 0.58}p_3$$
$$= p_1 + \frac{0.58 + \delta}{0.71 - \delta}p_2 + \frac{0.87}{0.42}p_3$$
$$= 1 + \frac{2\delta - 0.13}{0.71 - \delta}p_2 + \frac{15}{14}p_3$$
$$\geq 1 + \frac{2\delta - 0.13}{0.71 - \delta}p_2 + \frac{15}{14} \cdot \frac{0.21 - \delta - (0.29 - \delta)p_2}{0.71 - \delta}$$
$$= 1 + \left(\frac{2\delta - 0.13}{0.71 - \delta} - \frac{15}{14} \cdot \frac{0.29 - \delta}{0.71 - \delta}\right)p_2 + \frac{15}{14} \cdot \frac{0.21 - \delta}{0.71 - \delta}$$
$$\geq 1.12579,$$

where in the last inequality, we substitute $\delta = \frac{617}{4300}$ so that the term in front of $p_2$ becomes equal to zero. Note that $\mathrm{ER}(\mathrm{mid}(\mathbf{x}'), \mathbf{x}') = 1$, leading to our approximation ratio lower bound of $1.12579$.

For the remaining case where $\mathbb{E}_{y \in f(\mathbf{x})}[|y - 0.71|] \geq 0.21$, we let $\mathbf{x}' = (0.29, 1)$ and make a symmetric argument. $\qquad\square$

## C  Missing Details on Randomized Mechanisms with Prediction

For randomized mechanisms with prediction, by incorporating the newly devised $(\frac{\sqrt{5}}{2} - 1, \frac{2}{5})$-LRM constant mechanism with the learning augmentation scheme, we have the following randomized mechanism.

---

**Mechanism 4** $\alpha$-Bounding Interval Randomized Mechanism

---

**Input:** Location profile $\mathbf{x}$, optimal locaiton prediction $\hat{y}$.
**Output:** Distribution of facility location $f(\mathbf{x}, \hat{y})$.
 1: Initialize the confidence parameter $\alpha \in (1, 2]$.
 2: **if** $\hat{y} \in [1 - \frac{1}{\alpha}, \frac{1}{\alpha}]$ **then**
 3:     Return $f(\mathbf{x}, \hat{y}) \leftarrow \hat{y}$;
 4: **else**
 5:     Return $f(\mathbf{x}, \hat{y}) \leftarrow (\frac{\sqrt{5}}{2} - 1, \frac{2}{5})$-LRM constant mechanism with input $\mathbf{x}$.
 6: **end if**

---

The $\alpha$-Bounding Interval Randomized Mechanism adopts a similar approach to $\alpha$-BIM. It begins by defining a "trustworthy" interval associated with the predicted optimal facility location. When the prediction falls outside this interval, the mechanism employs the $(\frac{\sqrt{5}}{2} - 1, \frac{2}{5})$-LRM constant mechanism to ensure the small robustness. However, our next analysis shows that it achieves even worse consistency and robustness than $\alpha$-BIM.

**Proposition C.1.** *$\alpha$-Bounding Interval Randomized Mechanism is strategyproof and satisfies* $\min\{1 + (\frac{12 + 4\sqrt{5}}{5})(1 - \frac{1}{\alpha}), (\frac{3}{5} + \frac{2\sqrt{5}}{5}) + \frac{8}{5}(1 - \frac{1}{\alpha}), 1 + \frac{2}{\sqrt{5}}\}$-*consistency,* $\frac{\alpha}{\alpha - 1}$-*robustness with respect to envy ratio, where* $\alpha \in (1, 2]$.

*Proof.* Strategyproofness directly holds for the $\alpha$-Bounding Interval Randomized Mechanism. We mainly focus on the proof of the bounds of consistency and robustness.

**Consistency**. Consider any instance $\mathbf{x}$ with a correct prediction of the optimal facility location, i.e., $\hat{y} = \mathrm{mid}(\mathbf{x})$. If $\hat{y}$ is within $[1 - \frac{1}{\alpha}, \frac{1}{\alpha}]$, the mechanism outputs $\hat{y}$, providing 1-consistency. Next, we consider the case where $\hat{y} \in [0, 1 - \frac{1}{\alpha}]$, in which the mechanism returns the output of the $(\frac{\sqrt{5}}{2} - 1, \frac{2}{5})$-LRM constant mechanism. Here, we consider the following sub-cases divided by the range of the parameter $\alpha$. By Lemma 3.1, it suffices to consider instances with two agents where $0 \leq x_1 < x_2 \leq 2(1 - \frac{1}{\alpha})$.

We first notice that when $\alpha \in [\frac{4}{3}, 2]$, i.e., $2(1 - \frac{1}{\alpha}) \in [\frac{1}{2}, 1]$, the consistency is always bounded by $1 + \frac{2}{\sqrt{5}}$ since there always exists one two-agent instance achieving approximation ratio of $1 + \frac{2}{\sqrt{5}}$, regardless the prediction location $\hat{y}$. Therefore, we mainly consider two cases

**Case (1).** $2(1 - \frac{1}{\alpha}) \leq \frac{3 - \sqrt{5}}{2}$, i.e., $\alpha \in [1, \sqrt{5} - 1]$ which implies that both $x_1$ and $x_2$ are on the left side of $\frac{3 - \sqrt{5}}{2}$. Henceforth, the approximation ratio is upper-bounded by

$$\rho(\mathbf{x}) \leq \frac{2}{5} \cdot \frac{1 - (\frac{3 - \sqrt{5}}{2} - 2(1 - \frac{1}{\alpha}))}{1 - \frac{3 - \sqrt{5}}{2}} + \frac{1}{5} \cdot \frac{1 - (\frac{1}{2} - 2(1 - \frac{1}{\alpha}))}{1 - \frac{1}{2}} + \frac{2}{5} \cdot \frac{1 - (\frac{\sqrt{5} - 1}{2} - 2(1 - \frac{1}{\alpha}))}{1 - \frac{\sqrt{5} - 1}{2}}$$

$$= 1 + (\frac{12 + 4\sqrt{5}}{5})(1 - \frac{1}{\alpha}),$$

where the equality holds when $x_1 = 0$ and $x_2 = 2(1 - \frac{1}{\alpha})$.

**Case (2).** $2(1 - \frac{1}{\alpha}) > \frac{3 - \sqrt{5}}{2}$, i.e., $\alpha \in (\sqrt{5} - 1, \frac{4}{3}]$. The approximation ratio is upper-bounded by

$$\rho(\mathbf{x}) \leq \frac{2}{5} \cdot \frac{1 - (2(1 - \frac{1}{\alpha}) - \frac{3 - \sqrt{5}}{2})}{1 - \frac{3 - \sqrt{5}}{2}} + \frac{1}{5} \cdot \frac{1 - (\frac{1}{2} - 2(1 - \frac{1}{\alpha}))}{1 - \frac{1}{2}} + \frac{2}{5} \cdot \frac{1 - (\frac{\sqrt{5} - 1}{2} - 2(1 - \frac{1}{\alpha}))}{1 - \frac{\sqrt{5} - 1}{2}}$$

$$= (\frac{3}{5} + \frac{2\sqrt{5}}{5}) + \frac{8}{5}(1 - \frac{1}{\alpha}).$$

where the equality holds when $x_1 = 0$ and $x_2 = 2(1 - \frac{1}{\alpha})$.

**Robustness**. Now we consider the robustness of the mechanism. By Mechanism 1, we can see that **if** branch achieves $\frac{\alpha}{\alpha - 1}$-robustness and **else** branch achieves $\frac{21}{11} < \frac{\alpha}{\alpha - 1}$ robustness. Hence, the robustness $\beta = \frac{\alpha}{\alpha - 1}$. $\square$

Next, we extend BAM by leveraging the $(\frac{\sqrt{5}}{2} - 1, \frac{2}{5})$-LRM Mechanism, that is, with probability $(1 - p)$ running $(\frac{\sqrt{5}}{2} - 1, \frac{2}{5})$-LRM mechanism, rather than putting the facility at $\frac{1}{2}$.

---

**Mechanism 5** Bias-Aware LRM Mechanism

---

**Input:** Location profile $\mathbf{x}$, facility location prediction $\hat{y}$.
**Output:** Facility location $f(\mathbf{x}, \hat{y})$.
 1: Compute bias $c = |\hat{y} - \frac{1}{2}|$
 2: Compute probability $p = \frac{1}{2} - c$
 3: With probability $p$: return $\hat{y}$
 4: With probability $1 - p$: return $(\frac{\sqrt{5}}{2} - 1, \frac{2}{5})$-LRM constant mechanism with input $\mathbf{x}$.

---

We next show that Bias-Aware LRM Mechanism is worse than BAM by constructing some special instances.

**Proposition C.2.** *Bias-Aware LRM mechanism is strategyproof and satisfies* $\left(\frac{23 + 9\sqrt{5}}{20}\right)$-*consistency and* $\left(\frac{1}{2} + \frac{4}{\sqrt{5}}\right)$-*robustness when* $c \in [0, \frac{1}{4}]$, *and* $\left(\frac{-8c^2 + 2(\sqrt{5} - 1)c + \sqrt{5} + 6}{5}\right)$-*consistency,* $\left(\frac{1}{10}(4\sqrt{5}c + 7\sqrt{5} + 5)\right)$-*robustness when* $c \in [\frac{1}{4}, \frac{\sqrt{5} - 1}{4}]$, *and* $\left(-\frac{4(3 + \sqrt{5})}{5}c^2 + \frac{\sqrt{5} + 8}{5}\right)$-*consistency,* $\left(\frac{1}{10}(4\sqrt{5}c + 7\sqrt{5} + 5)\right)$-*robustness when* $c \in [\frac{\sqrt{5} - 1}{4}, \frac{1}{2}]$.

*Proof.* Strategyproofness directly holds for the Bias-Aware LRM mechanism. We mainly focus on the proof of the lower bounds of consistency and robustness. Without loss of generality, we assume that $\hat{y} \leq \frac{1}{2}$.

**Consistency**. By Lemma 3.1 and the proof of Theorem 4.3, we only need to consider instances with two agents where $0 = x_1 < x_2 \leq 1$. Note that the envy ratio achieved by $\hat{y}$ is 1. The envy ratio achieved by $(\frac{\sqrt{5}}{2} - 1, \frac{2}{5})$-LRM is always $\frac{5+2\sqrt{5}}{5}$. When moving $x_2$ from location 1 to location $\frac{\sqrt{5}-1}{2}$ (the right boundary of $(\frac{\sqrt{5}}{2} - 1, \frac{2}{5})$-LRM), the probability of using $(\frac{\sqrt{5}}{2} - 1, \frac{2}{5})$-LRM will increase. Thus the expected approximation ratio will increase. Therefore, we only need to consider the case where $0 = x_1 < x_2 \leq \frac{\sqrt{5}-1}{2}$. Consider any instance $\mathbf{x}$ with correct prediction of optimal facility location $\hat{y}$, i.e., $\mathrm{mid}(\mathbf{x}) = \hat{y}$. We have $\hat{y} \leq \frac{\sqrt{5}-1}{4}$.

If $0 \leq \hat{y} \leq \frac{3-\sqrt{5}}{4}$, then $c \in [\frac{\sqrt{5}-1}{4}, \frac{1}{2}]$, and the consistency is

$$
\begin{aligned}
\gamma &= \frac{\mathrm{ER}(f(\mathbf{x}, \hat{y}), \mathbf{x})}{\mathrm{ER}(\mathrm{mid}(\mathbf{x}), \mathbf{x})} \\
&= \hat{y} \cdot 1 + (1 - \hat{y}) \left( \frac{2}{5} \cdot \frac{\frac{\sqrt{5}-1}{2} + 2\hat{y}}{\frac{\sqrt{5}-1}{2}} + \frac{1}{5} \cdot \frac{\frac{1}{2} + 2\hat{y}}{\frac{1}{2}} + \frac{2}{5} \cdot \frac{\frac{3-\sqrt{5}}{2} + 2\hat{y}}{\frac{3-\sqrt{5}}{2}} \right) \\
&= -\frac{4(3 + \sqrt{5})}{5} c^2 + \frac{\sqrt{5} + 8}{5},
\end{aligned}
$$

which is monotonically decreasing with respect to $c$.

If $\frac{3-\sqrt{5}}{4} \leq \hat{y} \leq \frac{1}{4}$, then $c \in [\frac{1}{4}, \frac{\sqrt{5}-1}{4}]$, and the consistency is

$$
\begin{aligned}
\gamma &= \frac{\mathrm{ER}(f(\mathbf{x}, \hat{y}), \mathbf{x})}{\mathrm{ER}(\mathrm{mid}(\mathbf{x}), \mathbf{x})} \\
&= \hat{y} \cdot 1 + (1 - \hat{y}) \left( \frac{2}{5} \cdot \frac{\frac{5-\sqrt{5}}{2} - 2\hat{y}}{\frac{\sqrt{5}-1}{2}} + \frac{1}{5} \cdot \frac{\frac{1}{2} + 2\hat{y}}{\frac{1}{2}} + \frac{2}{5} \cdot \frac{\frac{3-\sqrt{5}}{2} + 2\hat{y}}{\frac{3-\sqrt{5}}{2}} \right) \\
&= \frac{1}{5} \left( -8c^2 + 2(\sqrt{5} - 1)c + \sqrt{5} + 6 \right)
\end{aligned}
$$

which is monotonically decreasing with respect to $c$.

If $\frac{1}{4} \leq \hat{y} \leq \frac{\sqrt{5}-1}{4}$, then $c \in [\frac{3-\sqrt{5}}{4}, \frac{1}{4}]$, and the consistency is

$$
\begin{aligned}
\gamma &= \frac{\mathrm{ER}(f(\mathbf{x}, \hat{y}), \mathbf{x})}{\mathrm{ER}(\mathrm{mid}(\mathbf{x}), \mathbf{x})} \\
&= \hat{y} \cdot 1 + (1 - \hat{y}) \left( \frac{2}{5} \cdot \frac{\frac{5-\sqrt{5}}{2} - 2\hat{y}}{\frac{\sqrt{5}-1}{2}} + \frac{1}{5} \cdot \frac{\frac{3}{2} - 2\hat{y}}{\frac{1}{2}} + \frac{2}{5} \cdot \frac{\frac{3-\sqrt{5}}{2} + 2\hat{y}}{\frac{3-\sqrt{5}}{2}} \right) \\
&= \frac{2c}{\sqrt{5}} + \frac{1}{\sqrt{5}} + 1,
\end{aligned}
$$

which is monotonically increasing with respect to $c$.

**Robustness**. If $0 \leq \hat{y} \leq \frac{3-\sqrt{5}}{2}$, then $c \in [\frac{\sqrt{5}-2}{2}, \frac{1}{2}]$. By using a similar analysis as Theorem 4.3 we have that the worst case satisfies $x_2 = 1$. If $x_1 < \hat{y}$, we can show that moving this agent from $x_1$ to $\hat{y}$ will increase the expected envy ratio. To see this, the envy ratio achieved by $\hat{y}$ is increasing and the envy ratio achieved by $(\frac{\sqrt{5}-1}{2}, \frac{2}{5})$-LRM constant mechanism is increasing. If $x_1 > \frac{3-\sqrt{5}}{2}$, we can also use a similar analysis to show that moving this agent from $x_1$ to $\frac{\sqrt{5}-1}{2}$ will increase the expected envy ratio. Then we consider $x_1 \in [\hat{y}, \frac{1}{3}]$. Let $\delta = x_1 - \hat{y}$, by using the similar analysis as Theorem 4.3 we can show that the approximation ratio satisfies the monotonicity with respect to $\delta$. Hence, we only need to compare the envy ratio between two cases $(\hat{y}, 1)$ and $(\frac{3-\sqrt{5}}{2}, 1)$.

For case $(\hat{y}, 1)$, if $\hat{y} \le \sqrt{5} - 2$ ($c \in [\frac{5}{2} - \sqrt{5}, \frac{1}{2}]$), we have the robustness

$$\beta = \hat{y} \cdot \frac{1}{\hat{y}} + (1 - \hat{y}) \left( \frac{2}{5} \cdot \frac{\frac{\sqrt{5}-1}{2} + \hat{y}}{\frac{3-\sqrt{5}}{2}} + \frac{1}{5} \cdot \frac{\frac{1}{2} + \hat{y}}{\frac{1}{2}} + \frac{2}{5} \cdot \frac{\frac{\sqrt{5}-1}{2}}{\frac{3-\sqrt{5}}{2} + \hat{y}} \right)$$

$$= -\frac{4(\sqrt{5} - 5)c^3 + 4(4\sqrt{5} - 13)c^2 + (21\sqrt{5} - 53)c - 34\sqrt{5} + 89}{5(\sqrt{5} - 3)(2c + \sqrt{5} - 4)}.$$

If $\sqrt{5} - 2 \le \hat{y} \le \frac{3-\sqrt{5}}{2}$ ($c \in [\frac{\sqrt{5}-2}{2}, \frac{5}{2} - \sqrt{5}]$), we have the robustness

$$\beta = \hat{y} \cdot \frac{1}{\hat{y}} + (1 - \hat{y}) \left( \frac{2}{5} \cdot \frac{\frac{\sqrt{5}-1}{2} + \hat{y}}{\frac{3-\sqrt{5}}{2}} + \frac{1}{5} \cdot \frac{\frac{1}{2} + \hat{y}}{\frac{1}{2}} + \frac{2}{5} \cdot \frac{\frac{3-\sqrt{5}}{2} + \hat{y}}{\frac{\sqrt{5}-1}{2}} \right)$$

$$= \frac{1}{10} \left( -4(3 + \sqrt{5})c^2 + (2 + 4\sqrt{5})c + 3\sqrt{5} + 14 \right).$$

For case $(\frac{3-\sqrt{5}}{2}, 1)$, we have the robustness

$$\beta = \hat{y} \cdot \frac{\frac{\sqrt{5}-1}{2} + \hat{y}}{\hat{y}} + (1 - \hat{y}) \left( \frac{2}{\sqrt{5}} + 1 \right)$$

$$= \frac{1}{10}(4\sqrt{5}c + 7\sqrt{5} + 5),$$

which is monotonically increasing with respect to $c$, and always larger than the robustness achieved by $(\hat{y}, 1)$.

If $\frac{3-\sqrt{5}}{2} \le \hat{y} \le \frac{1}{2}$, then $c \in [0, \frac{\sqrt{5}-2}{2}]$. When $x_1 = \hat{y}$ and $x_2 = 0$, the envy ratios achieved by $\hat{y}$ and $(\frac{\sqrt{5}-1}{2}, \frac{2}{5})$-LRM reach the maximum. Then the robustness is

$$\beta = \hat{y} \cdot \frac{1}{\hat{y}} + (1 - \hat{y}) \cdot (\frac{5 + 2\sqrt{5}}{5}))$$

$$\le (\frac{2}{\sqrt{5}} + 1)c + \frac{1}{\sqrt{5}} + \frac{3}{2},$$

which is monotonically increasing with respect to $c$. Combined with the consistency, we have that when $c \in [0, \frac{1}{4})$, both the consistency and robustness are better than $c = \frac{1}{4}$ (in this case, it is upper-bounded by $(\frac{23+9\sqrt{5}}{20})$-consistency and $(\frac{1}{2} + \frac{4}{\sqrt{5}})$-robustness), which can be omitted. □

We compare the consistency and robustness of all aforementioned four mechanisms, including $\alpha$-BIM, BAM, $\alpha$-Bounding Interval Randomized Mechanism, and Bias-Aware LRM Mechanism in Figure 5.

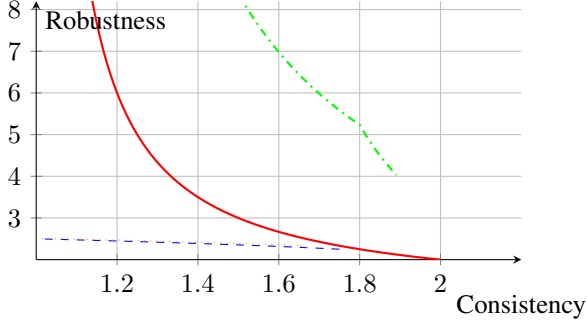

Figure 5: Comparison between $\alpha$-BIM (red solid line), BAM (blue dashed line), $\alpha$-Bounding Interval Randomized Mechanism (green dashdotted line), and Bias-Aware LRM Mechanism (orange dotted line).

BAM clearly outperforms both $\alpha$-BIM and the $\alpha$-Bounding Interval Randomized Mechanism. While the Bias-Aware LRM Mechanism shares the same framework as BAM, it is slightly less effective and involves greater complexity.

