# OpenReview forum: "Learning-Augmented Facility Location Mechanisms for Envy Ratio"
_NeurIPS.cc/2025/Conference — NeurIPS 2025 poster_

### Official Review · Reviewer_Lmi3 · 2025-06-07

**Clarity:** 4
**Significance:** 2
**Originality:** 2
**Rating:** 4
**Confidence:** 3

**Summary:**

This paper studies a learning-augmented version of the facilitation location mechanism design problem: a variant with the ratio envy ratio objective, which is the max ratio of utilities of any two agents.

For deterministic mechanisms: they give a $(\alpha,\frac{\alpha}{\alpha-1})$ consistency-robustness tradeoff.

For randomized mechanisms:

For the setting without predictions,
they  show an improvement in the upper bound (from $2$ to approximately $1.89$) and in the lower bound (from approximately $1.03$ to approximately $1.11$).

For the setting with predictions, they propose the BAM randomized mechanism that gets $(-4c^2 + 2, c+2)$-consistency-robustness trade-off for $c \in [0.25,0.5]$ and $(\frac{7}{4}, \frac{7}{4})$ for $c \in [0,0.25)$, where $c$ is the deviation from the interval midpoint $\hat{y}$. They investigate other
 potential mechanisms as well.

In the paper they give 3 mechanisms $BIM$, $LRM$ and $BAM$:
The $BIM$ mechanism is essentially a one dimensional predetermined hard-coded bounding-box mechanism, which chooses the prediction if it is inside the box or the closest box endpoint otherwise. The $LRM$ mechanism roughly returns the hard-coded middle point of the assumed segment or a random point in a constant distance from it. The $BAM$ mechanism returns the middle or the prediction according to a coin-flip determined by the distance between the prediction and the middle point.

**Questions:**

* Why is the restriction to an interval needed in the setting? I am aware that Ding [2020] also have this assumption. Are there any results for the general setting (locations are some real numbers)?

* What is the motivation for defining the problem (again, as Ding [2020]) as a utility maximization problem rather than a cost minimization one?
    Do the mechanisms fail otherwise?

* Line 224: "By Moulin's [1980] characterization...": Could you point to the exact place in Moulin's paper?

* Line 312: Could you add why an explanation for why if $f(x,\hat{y}) = 0.5$ the envy ratio is the one written?

Small possible fixes:
* Line 293: I would replace "To summarize" with "To demonstrate".
* Line 310: I would change the order to: Let $\delta = \frac{x_2 - x_1}{2}$ rather than the other way around.

**Ethical Concerns:**

["NO or VERY MINOR ethics concerns only"]

**Final Justification:**

The techniques used to achieve the consistency–robustness results are fairly standard and relatively straightforward.

All proposed mechanisms—$BIM$, $LRM$, and $BAM$—are trivially strategyproof and anonymous, as they rely solely on the prediction and not on any input reported by the agents. This effectively reduces the problem from a mechanism design setting to the much simpler task of approximating the midpoint of a sub-segment of a known segment, where the sub-segment itself is unknown.

In my view, this is the paper’s main weakness and the primary reason why my score remains at borderline accept rather than higher.

**Limitations:**

yes

**Paper Formatting Concerns:**

Yes

**Quality:**

3

**Strengths And Weaknesses:**

Strengths:

* This work improves the best known lower and upper bounds on the approximation ratio obtained by random strategyproof mechanisms: The upper bound is improved from $2$ to $\approx 1.89$ and the lower bound is improved from $1.03$ to $\approx 1.126$.

* The paper provides consistency-robustness tradeoffs for their mechanism.

* It is interesting that more can be done without using the agent reported locations.

* The prediction settings results demonstrate a reasonable smooth consistency-robustness tradeoff. The randomized algorithm achieves a better trade-off than the deterministic one.

* I did not find any errors in the proofs.

Weaknesses:
* Usually, the main challenge in mechanism design is to ensure strategyproofness (or some variant of it).
    In mechanism design with predictions, usually the challenge is to try and make use of the input together with the predictions to come up with a strategy-proof mechanism with a good approximation ratio.

* All given mechanisms, $BIM$, $LRM$ and $BAM$, are trivially strategyproof and anonymous as they do not use the input (reported by the agents) at all, but only the prediction. This reduces the problem from a mechanism design problem to the much simpler problem of approximating the midpoint of a sub-segment of a given known segment, where the sub-segment is not known.
I would encourage the authors to try and make use of the input reported by the agents as well. If the input does not help, specifying this would also benefit the paper.

* The techniques for achieving the consistency-robustness results are quite standard and rather straightforward: The $\alpha$-BIM mechanism is very similar to the minimum bounding box mechanism of Agrawal et al. 2022 (except it is even simpler as it does not use the agent locations but only hard-coded box-constraints). The control over the consistency-robustness trade-off boils down to deciding on the box endpoints. The BAM mechanism returns the prediction w.p. $p$ and the "hard-coded" segment midpoint otherwise, interpolating between $p=1$ (deterministically choosing the prediction) and $p=0$ (deterministically choosing the midpoint).

* A smaller weakness: Often in algorithms with predictions a confidence parameter is available to choose, roughly translating how much the algorithm/mechanism user trusts the predictions. In BAM there is no confidence parameter. The mechanism user can not control the exact location on the consistency-robustness tradeoff lines presented in Figure 2. We note this is not true for the $\alpha$-BAM mechanism.

---

> ### Author Rebuttal · Authors · 2025-07-30
>
> Thank you for your insightful comments and suggestions. We respond to your questions and concerns as follows.
>
> **Q: Why the restriction to an interval needed in the setting? Why define the problem as a utility maximization problem rather than cost minimization?**
>
> A: This setting, where agents and facilities are positioned on a fixed interval of the real line, is standard in the study of utility-based facility location problems, as seen in works such as “Heterogeneous Facility Location with Limited Resources” (GEB 2023) and “Facility Location Games with Fractional Preferences” (AAAI 2018). For additional references, see the survey “Mechanism Design for Facility Location Problems: A Survey” (IJCAI 2021). Our model employs a utility-based formulation for two key reasons: (1) it aligns with established definitions of envy ratio in prior work, and (2) a distance-based envy ratio results in the nonexistence of strategyproof mechanisms with bounded approximation ratios, leading to unbounded robustness. We also discussed it in the footnote on page 3 of our paper, but we will enhance these explanations in the revised version to ensure clarity.
>
> **Q: Line 224, "By Moulin's [1980] characterization...": Could you point to the exact place in Moulin's paper?**
>
> A: The characterization by Moulin’s [1980] is presented in Proposition 2. We will specify it in the updated version of the paper.
>
> **Q: Line 312, why if $f(x, \hat{y})=0.5$ the envy ratio is the one written?**
>
> A: Recall that $x_2 - x_1 = 2\delta$. Since we are analyzing consistency, $\hat{y}$ is the optimal solution, i.e., $\hat{y}$ is the midpoint of $x_1$ and $x_2$, so we have $x_2=\hat{y} + \delta$ and $x_1=\hat{y}-\delta$. Additionally, we consider the case $x_1 < x_2 < \frac{1}{2}$, thus $x_2$ achieves the maximum utility and $x_1$ achieves the minimum utility. Under these conditions, the envy ratio when locating the facility at $\frac{1}{2}$ is given by $$\frac{1-(\frac{1}{2}-x_2)}{1-(\frac{1}{2}-x_1)}.$$
>
> **Q: No confidence parameter for the BAM mechanism.**
>
> A: In contrast to $\alpha$-BIM, BAM is not parameterized and thus operates independently of any parameters, including confidence parameters. We have also explored randomized mechanisms that incorporate confidence parameters, allowing users to exert control; this is detailed in the $\alpha$-Bounding Interval Randomized Mechanism (see Appendix C). As shown in Figure 4, none of these alternative mechanisms outperform BAM. Nevertheless, we agree with the reviewer that mechanisms incorporating confidence parameters are desirable, provided they maintain strong approximation guarantees.
>
> **Q: Why considering agents’ reported locations does not help the mechanism to achieve a better approximation ratio? That is, why constant mechanism?**
>
> A: We provide an intuitive example illustrating that incorporating agents’ reported locations does not improve the outcome. Suppose there is one agent located at 0 and n−1 agents located at 1. The optimal facility location is at $\frac{1}{2}​$, yielding an envy ratio of 1. However, if the mechanism depends solely on the agents’ reported locations, the facility would be placed at either 0 or 1, resulting in an unbounded envy ratio. Consequently, the approximation ratio also becomes unbounded.
>
> **Q: The techniques for achieving the consistency-robustness results are quite standard and rather straightforward, i.e., not too many novel techniques.**
>
> A: We argue that most of our techniques are novel and provide insights for future works along this line of research.
> **Deterministic Mechanisms with Prediction.** Although our deterministic mechanism shares similarities with the seminal work of Agrawal et al. [2022], our proofs diverge significantly. Unlike their maximum cost objective, which does not involve utility functions in the denominator, our envy ratio incorporates utility functions in both the numerator and denominator, resulting in a more complex analytical structure. This complexity introduces substantial challenges in handling cases involving both maximum and minimum utilities. To address these difficulties, we introduce key lemmata, such as Lemma 3.1, that streamline and simplify the overall proof process.
>
> **Randomized Mechanisms Without Prediction.** We contend that both the design and analysis techniques for randomized mechanisms are novel and non-trivial. Our approximation analysis for randomized mechanisms without prediction is new and specifically tailored to the unique structure of the envy ratio objective. The progression of results from Lemma 4.1 to Theorem 4.3 reflects considerable technical effort and offers original insights to the literature.
>
> **Randomized Mechanisms With Prediction.** Our proposed Biased-Aware Mechanism (BAM) introduces a fundamentally new approach: the placement probability dynamically adjusts based on the prediction bias, defined as the distance between the predicted location and the midpoint $\frac{1}{2}$​. In contrast, existing literature typically employs fixed probabilities to combine different deterministic mechanisms. We also presented similar mechanisms in the appendix, but their performance is significantly inferior to that of BAM. The key intuition of BAM  is that when the prediction is near the endpoints of the interval, robustness requires the mechanism to drastically reduce the probability of placing the facility at the predicted location. This strategy avoids an unbounded approximation ratio in cases where the prediction is inaccurate. This idea is central to achieving the $\frac{7}{4}$-consistency result without significantly sacrificing robustness.
>
> We believe that our mechanism design and analytical techniques for the randomized setting provide novel contributions and offer valuable insights for future research in this area.
>
> **Q: Minor Comments.**
>
> A: We are grateful to the reviewer for pointing out these minor typos. We will correct them in the revised version of our paper.

---

> ### Comment · Reviewer_Lmi3 · 2025-08-02
>
> I thank the authors for their response.
>
> Followup:
>
> > A: This setting, where agents and facilities are positioned on a fixed interval of the real line, is standard in the study of utility-based facility location problems, as seen in works such as ... Our model employs a utility-based formulation for two key reasons: (1) it aligns with established definitions of envy ratio in prior work, and (2) a distance-based envy ratio results in the nonexistence of strategyproof mechanisms with bounded approximation ratios, leading to unbounded robustness. We also discussed it in the footnote on page 3 of our paper, but we will enhance these explanations in the revised version to ensure clarity.
>
> Thank you for the helpful references. As other reviewers have noted, the assumption that agents lie on a fixed interval plays a critical role in shaping the problem and the complexity of its solutions. Adding the reference helps.
>
> > A: We argue that most of our techniques are novel and provide insights for future works along this line of research. Deterministic Mechanisms with Prediction. Although our deterministic mechanism shares similarities with the seminal work of Agrawal et al. [2022], our proofs diverge significantly. Unlike their maximum cost objective, which does not involve utility functions in the denominator, our envy ratio incorporates utility functions in both the numerator and denominator, resulting in a more complex analytical structure. This complexity introduces substantial challenges in handling cases involving both maximum and minimum utilities. To address these difficulties, we introduce key lemmata, such as Lemma 3.1, that streamline and simplify the overall proof process.
> > Randomized Mechanisms Without Prediction. We contend that both the design and analysis techniques for randomized mechanisms are novel and non-trivial. Our approximation analysis for randomized mechanisms without prediction is new and specifically tailored to the unique structure of the envy ratio objective. The progression of results from Lemma 4.1 to Theorem 4.3 reflects considerable technical effort and offers original insights to the literature.
>
>
> While your response addresses the complexity of the analysis and the technical steps in the proofs, it does not fully address the concern regarding the novelty of the mechanism itself—particularly how the predictions are combined with standard mechanisms.
>
> For example, Lemma 3.1 reduces the analysis to two-agent instances, but this simplification is fairly direct given that envy is only affected by extreme points. As for Theorem 4.3, the mechanism's strategyproofness and anonymity follow almost immediately, since the agents' reports are unused. There is no intricate incentive-alignment technique involved.
>
> The approximation analysis, though careful, appears to follow standard techniques for analyzing multivariate rational functions. While it’s true that incorporating utility functions in both numerator and denominator complicates the expressions, this structure arguably detracts from the simplicity and interpretability of the model. It remains unclear whether the analysis techniques are general or tailored to this particular objective.
>
> > A: We provide an intuitive example illustrating that incorporating agents’ reported locations does not improve the outcome. Suppose there is one agent located at 0 and n−1 agents located at 1. The optimal facility location is at 0.5, yielding an envy ratio of 1. However, if the mechanism depends solely on the agents’ reported locations, the facility would be placed at either 0 or 1, resulting in an unbounded envy ratio. Consequently, the approximation ratio also becomes unbounded.
>
> This example shows that selecting only from reported locations can lead to arbitrarily bad outcomes. However, it does not rule out the possibility that using agent reports—together with predictions—could enable better approximation ratios. The example supports the limitations of restricted mechanisms, but not necessarily the impossibility of leveraging agent data more generally.
>
> > Randomized Mechanisms With Prediction. Our proposed Biased-Aware Mechanism (BAM) introduces a fundamentally new approach: the placement probability dynamically adjusts based on the prediction bias, defined as the distance between the predicted location and the midpoint . In contrast, existing literature typically employs fixed probabilities to combine different deterministic mechanisms. ...
>
> I agree that dynamically adjusting placement probabilities based on the bias between the predicted and standard solutions is an elegant idea.

---

> > ### Author Response · Authors · 2025-08-03
> >
> > We are very grateful to the reviewer for the detailed response to our rebuttal!
> >
> > > While your response addresses the complexity of the analysis and the technical steps in the proofs, it does not fully address the concern regarding the novelty of the mechanism itself—particularly how the predictions are combined with standard mechanisms.
> >
> > > For example, Lemma 3.1 reduces the analysis to two-agent instances, but this simplification is fairly direct given that envy is only affected by extreme points. As for Theorem 4.3, the mechanism's strategyproofness and anonymity follow almost immediately, since the agents' reports are unused. There is no intricate incentive-alignment technique involved.
> >
> > > The approximation analysis, though careful, appears to follow standard techniques for analyzing multivariate rational functions. While it’s true that incorporating utility functions in both numerator and denominator complicates the expressions, this structure arguably detracts from the simplicity and interpretability of the model. It remains unclear whether the analysis techniques are general or tailored to this particular objective.
> >
> > **(Novelty Concern)** Thank you for your response. We apologize that our earlier reply did not fully address your concern. We understand your concern regarding the novelty of our mechanisms. It is true that our $\alpha$-BIM mechanism, while inspired by the work of Agrawal et al. [2022], may offer limited novelty in design. However, $\alpha$-BIM achieves tight approximation bounds for our problem setting. In this sense, we believe it contributes meaningfully by adapting and extending existing elegant mechanisms to a new context with provably tight results, a contribution we consider non-negligible.
> >
> > Regarding randomized mechanisms, we have shown in our paper that simply following the ideas from prior work fails to yield good approximation guarantees in our setting (see Appendix C for details). This limitation motivated the development of our novel BAM mechanism. While we have tailored our analysis techniques to the envy ratio objective, we believe that similar approaches can be applied to a more-general space of objective functions. We are very grateful for the reviewer’s recognition of our design and believe that BAM offers a new perspective that may inspire future work on learning-augmented mechanism design.
> >
> > ---
> >
> > > This example shows that selecting only from reported locations can lead to arbitrarily bad outcomes. However, it does not rule out the possibility that using agent reports—together with predictions—could enable better approximation ratios. The example supports the limitations of restricted mechanisms, but not necessarily the impossibility of leveraging agent data more generally.
> >
> > **(Agent Location with Prediction)** Thank you for your follow-up! You are correct. The example we provided illustrates that selecting a facility location purely from reported agent positions can lead to arbitrarily bad approximation ratios, but it does not fully rule out the possibility of using agent reports in a more refined way. Regarding this point, we also made substantial efforts to explore alternative frameworks for randomized mechanisms. However, we were unable to identify any approach that outperforms our current design (see Appendix C for details). Our current conjecture is that directly incorporating agent reports into learning-augmented mechanisms (i.e., the mechanism places the facility on g(x,y) with positive probability, where g(x,y) is some arbitrary function of the reported locations and the prediction), either violates anonymity, strategyproofness, or gives worse consistency and robustness guarantees. Most notably, if the mechanism incorporates the reported locations in this case, we are still susceptible to unbounded approximation ratios, as illustrated in our earlier example. However, given the large space of randomized mechanisms which may satisfy these properties, we find that this is extremely difficult to prove.
> >
> > Since the bound for randomized mechanisms is still not tight, we also wanted to share additional insights that arose during the rebuttal process. In light of your comment and Reviewer jQBn’s suggestion, we believe there is a promising direction in incorporating prediction models with agents’ locations. Specifically, rather than predicting the optimal facility location, it may be more fruitful to consider predictions on the locations of the leftmost and rightmost agents, as it allows for more fine-grained analysis of consistency and robustness.
> >
> > We hope our discussion clarifies our current perspective and welcome any further suggestions you might have.

---

> > > ### Comment · Reviewer_Lmi3 · 2025-08-09
> > >
> > > thank the authors for their response. In my view, the concerns raised in my previous comment largely remain.
> > >
> > > I recommend adding a discussion on the use of agent locations in the paper. Omitting agent locations removes the incentive-based, game-theoretic aspect of the work, and therefore requires strong justification. You could also present your conjecture on this matter, explain why you believe proving it is challenging, and suggest it as a potential avenue for future research.
> > >
> > > Acknowledging the idea of dynamically adjusting placement probabilities based on the bias between the predicted and standard solutions, I will slightly raise my score.
> > >
> > > I have no further questions.

---

### Official Review · Reviewer_b8Qg · 2025-06-21

**Clarity:** 3
**Significance:** 2
**Originality:** 3
**Rating:** 4
**Confidence:** 4

**Summary:**

This paper studies the facility location problem, where all agents are located on $[0, 1]$, and a facility is chosen to serve them. The utility of an agent is $1$ minus her distance to the facility. Agents will misreport their locations in order to maximize their utilities, and the goal is to design a strategyproof mechanism with as small envy ratio (ER) as possible, where ER is defined as the ratio between the utilitys of the best-off agent and the worst-off agent.

This paper first gives a learning-augmented anonymous deterministic mechanism that is $\alpha$-consistent and $\frac{\alpha}{\alpha - 1}$-robust for a tunable parameter $\alpha \in [1, 2]$. Then, it is shown that no anonymous, deterministic, and strategyproof mechanism can strictly improve this consistency-robustness trade-off.

This paper also presents an anonymous randomized mechanism with an approximation ratio of $1.8944$, which strictly improves the best known approximation ratio of $2$ in prior work for randomized mechanisms without predictions. Then, it gives a learning-augmented anonymous randomized mechanism, whose approximation ratio depends on the bias of the prediction and strictly outperforms the deterministic one.

**Questions:**

- Is it possible to quantify the approximation ratio of the learning augmented mechanisms in terms of the error of the prediction?

- For randomized mechanisms, ex-ante strategyproofness is considered in the paper. Can similar approximation ratio guarantees be achieved when requiring ex-post strategyproofness, i.e., the mechanism is still strategyproof after fixing the randomness?

- Line 236: It is claimed that the facility will be placed at $1/\alpha$. However, it seems to me that it can also be placed at a location strictly smaller than $1/\alpha$ since it is possible that the locations of all phantoms are strictly smaller than $1/\alpha$. Could you address this or point out if I'm wrong?

**Ethical Concerns:**

["NO or VERY MINOR ethics concerns only"]

**Final Justification:**

My main complaint, which concerns about the smoothness guarantee, has been mostly addressed. Although as Reviewer jQBn pointed out, the proof is not included in the submission so the correctness of the claim is hard to verify, it sounds quite plausible to me, and I believe it's correct. Altogether, I think this submission slightly exceeds the bar of NeurIPS, as reflected in my score raise.

**Limitations:**

Yes.

**Paper Formatting Concerns:**

No.

**Quality:**

2

**Strengths And Weaknesses:**

As both facility location and learning-augmented mechanism design have been trendy research topics, the problem considered in this paper is interesting and well-motivated. The results of this paper are complete and non-trivial, and they strictly improve those in prior work. Moreover, this paper is clearly written, and I enjoy reading it.

Personally, I find the assumption that the metric space is a line quite restricted, although this may be a common assumption in the facility location literature. While the learning-augmented mechanisms given in the paper achieve optimal or improved consistency-robustness guarantees, one complaint I have is that no smoothness guarantee is provided. Namely, it is unclear whether the mechanisms admit an improved approximation ratio compared to robustness when the prediction is only nearly perfect. This is, from my perspective, crucial in this setting since in a continuous metric space, one cannot hope to obtain a perfect prediction, rendering the consistency guarantee less useful.

In addition, for the randomized learning-augmented mechanism, although it admits desirable expected guarantees, it seems to me that these guarantees have a large variance: even when the prediction is very close to $0$ or $1$, it has a small but not negligible probability to be chosen, in which case the approximation ratio will become very large. In this sense, the deterministic mechanism is arguably more appealing.

Detailed comments:
- The assumption that the metric space is a line should be stated in the abstract as well.
- It should be stated in the introduction that the consistency-robustness guarantee of the randomized mechanism is strictly better than that of the deterministic one since the comparison is not straightforward.
- Line 225-226: The sentence "This mechanism takes the..." is hard to parse.
- Line 264: It is better to be explicit about the dependence on $\alpha$ in the notation $x'$.
- Line 266: The notation $\rho(x)$ is not defined.

---

> ### Author Rebuttal · Authors · 2025-07-30
>
> Thank you for your insightful comments and suggestions. We respond to your questions and concerns as follows.
>
> **Q: Quantify the approximation ratio in terms of the prediction error.**
>
> A: Thank you for raising the issue regarding smoothness analysis. We initially derived results for the approximation ratio with respect to prediction error. However, we chose not to include them in the main text, as their analysis is sufficiently complex and the results are less intuitive than the (consistency, robustness) framework. In the revised version, we will incorporate these results, or at least include a related discussion, to provide insights for interested readers.
>
> For the deterministic mechanism with prediction, $\alpha$-BIM,  let $\eta$ be the prediction error. Notice that the mechanism $\alpha$-BIM is parameterized by $\alpha$, We have the following approximation ratio result.
> When $\alpha \leq \frac{1+\sqrt{5}}{2}$, we have the approximation ratio
> $$
> \rho_{\alpha}(\eta)=
> \begin{cases}
> \alpha & \eta \in [0, \frac{\alpha - 1}{2(\alpha + 1)}]\\\\
> 1 + \frac{4\eta}{1-2\eta} & \eta \in [\frac{\alpha - 1}{2(\alpha + 1)}, \frac{1}{\alpha}-\frac{1}{2}]\\\\
> 1 + \frac{2\alpha \eta}{\alpha - 1} & \eta \in [\frac{1}{\alpha}-\frac{1}{2}, \frac{1}{2\alpha}] \\\\
> \end{cases}
> $$
> When $\alpha > \frac{1+\sqrt{5}}{2}$, we have the approximation ratio
> $$
> \rho_{\alpha}(\eta)=
> \begin{cases}
> \alpha & \eta \in [0, \frac{(\alpha - 1)^2}{2\alpha}]\\\\
> 1 + \frac{2\alpha\eta}{\alpha - 1} & \eta \in [ \frac{(\alpha - 1)^2}{2\alpha}, \frac{1}{2\alpha}]\\\\
> \end{cases}
> $$
> Intuitively, Assume that we know the prediction model has an error smaller than $\frac{\sqrt{5}}{2} - 1\approx 0.118$, then we have the approximation ratio of $\frac{\sqrt{5}+1}{2}\approx 1.62$. **This implies that when we have a good prediction model with bounded small error, we can significantly improve the approximation ratio from 2 to 1.62 by our proposed $\alpha$-BIM.**
>
> For the randomized mechanism, unfortunately, providing an explicit expression for $\rho_{\alpha}(\eta)$ is indeed difficult, and we outline the reasons below. Similar to the smooth analysis of deterministic mechanisms, the randomized one also involves case-by-case examination based on the ranges of $\hat{y}$ and $\eta$. Each case is expressed in the form $\max_j { f_j(\eta, \hat{y}) }$. However, unlike deterministic mechanisms, the presence of $\hat{y}$ in the probability terms introduces significant complexity. Specifically, the numerator of most $f_j(\eta, \hat{y})$ terms includes quadratic expressions like $\hat{y}^2$ and cross terms like $\hat{y} \cdot \eta$, while the denominator involves both $\hat{y}$ and $\eta$. This inherent complexity prevents us from deriving a closed-form expression for $\rho_{\alpha}(\eta)$.
>
> **Q: Ex-post strategyproofness guarantee for randomized mechanisms.**
>
> A: It is worth noting that our randomized mechanism is constructed from three deterministic constant mechanisms, each of which directly satisfies ex-post strategyproofness.
>
> **Q: Line 236, the lower bound proof of the deterministic mechanism with prediction. The facility can be placed at a location strictly smaller than $\frac{1}{\alpha}$.**
>
> A: Yes, you are correct. It should say that the facility must be placed in $[1-\frac{1}{\alpha},\frac{1}{\alpha}]$. This still leads to alpha-consistency at best, so the correctness of the proof is unchanged here. Thank you for spotting this error, we will fix it in the updated version of the paper.
>
> **Q: (Weakness) For the randomized mechanism with prediction. It is still possible to have a bad approximation ratio when prediction lies closely to 0 or 1.**
>
> A: The dynamic probability approach achieves a low expected approximation ratio, as illustrated in Figure 2, particularly when the prediction lies near 0 or 1. This effectively mitigates the large envy ratio that would result from placing the facility exactly at the predicted location (mentioned by the reviewer), which is a key feature of our Biased-Aware Mechanism (BAM). Specifically, when the prediction is close to the endpoints of the interval, the mechanism significantly reduces the probability of placing the facility at the predicted location to maintain robustness. This reduction is crucial to avoid an unbounded approximation ratio in cases where the prediction is inaccurate.
>
> **Q: Minor comments.**
>
> A: We are grateful to the reviewer for pointing out these minor typos. We will correct them in the revised version of our paper.

---

> > ### Comment · Reviewer_b8Qg · 2025-08-02
> >
> > I thank the authors for your response. I think my questions and concerns are sufficiently addressed, and I decide to raise my score. Looking forward to seeing the revised version of the paper, particularly with the smoothness analysis.

---

### Official Review · Reviewer_gHpJ · 2025-06-27

**Clarity:** 4
**Significance:** 3
**Originality:** 3
**Rating:** 5
**Confidence:** 4

**Summary:**

This paper studies the design of strategyproof mechanisms for the facility location problem on the interval $[0,1]$, focusing on the envy ratio objective. The authors explore this problem within the learning-augmented framework, where the algorithm is given a potentially inaccurate prediction of the optimal facility location. The objective is to design algorithms with good consistency-robustness tradeoffs.

The paper first introduces a new deterministic mechanism called $\alpha$-Bounding Interval Mechanism ($\alpha$-BIM), which uses a parameter $\alpha \in [1,2]$ to tune the tradeoff between consistency and robustness. The authors prove that $\alpha$-BIM achieves optimal trade-offs between these two measures for deterministic strategyproof and anonymous mechanisms.

In the randomized setting without predictions, the authors construct a randomized strategyproof mechanism that achieves an approximation ratio of $1+2/\sqrt{5}$, improving on the previous best-known bound of 2. They also establish a lower bound of 1.1125 on the approximation ratio of any randomized mechanism, improving also upon the best known lower bound of 1.0314.

Finally, they propose the Biased-Aware Mechanism (BAM) for the randomized setting with prediction, achieving a consistency-robustness tradeoff strictly better than $\alpha$-BIM, which they proved to be optimal for deterministic algorithms.

**Questions:**

- Do the authors think that their algorithms have smoothness guarantees that they did not explore? Or are the algorithms brittle? (See for example "Pareto-Optimality, Smoothness, and Stochasticity in Learning-Augmented One-Max-Search" for the definition of brittleness). In any case, I hope the authors will provide some evidence or intuition to support their answer.
- In the randomized setting without predictions, the gap between the upper and lower bounds is still big. Do the authors have some intuition on possible future directions to narrow it more? And if the improvement should be significant on the upper or the lower bound?

**Ethical Concerns:**

["NO or VERY MINOR ethics concerns only"]

**Final Justification:**

I maintain my score of 5, as I believe the paper studies an interesting problem and proves strong results.
The authors addressed my question regarding the smoothness of the deterministic algorithm. Although I am not sure of the technical details behind their smoothness claim in the rebuttal, it seems correct, and it should not be very difficult to derive.

**Limitations:**

yes

**Quality:**

3

**Strengths And Weaknesses:**

### **Strengths**
- The problem studied is an interesting fundamental problem
- The results of the paper cover deterministic and randomized algorithms, with and without predictions.
- The consistency-robustness tradeoff for deterministic algorithms is tight.
- The gap between upper and lower bounds in the randomized setting (without predictions) is narrowed by improving both the upper and lower bound, which is a very nice contribution.
- The paper is very well written and easy to follow.

### **Weaknesses**
- In the learning-augmented framework, the authors focus on consistency and robustness, but do not study the smoothness of their algorithms, which is crucial criterion for algorithms with predictions, characterizing how the algorithm performance degrades as the prediction error increases.
- While $\alpha$-BIM is well-explained, there isn't much intuition on how the algorithm was designed.

### **Comments**
- In lines 85, 304 and 317, the paper states that BAM achieves a consistency of 7/4 and a robustness of 7/4. I believe this is a typo. Following the proof, the robustness achieved by BAM is 9/4, which equals 2.25 as shown in Figure 2. I trust the authors will correct this.

---

> ### Author Rebuttal · Authors · 2025-07-30
>
> Thank you for your insightful comments and suggestions. We respond to your questions and concerns as follows.
>
> **Q: Smoothness analysis for the proposed algorithms.**
>
> A: Thank you for raising the issue regarding smoothness analysis. We argue that the $\alpha$-BIM satisfies smoothness as the approximation ratio is a **continuous** and **monotonic** function of the prediction error. The supporting evidence is as follows. Let $\eta$ be the prediction error. Notice that the mechanism $\alpha$-BIM is parameterized by $\alpha$, We have the following approximation ratio result. When $\alpha \leq \frac{1+\sqrt{5}}{2}$, we have the approximation ratio
> $$
> \rho_{\alpha}(\eta)=
> \begin{cases}
> \alpha & \eta \in [0, \frac{\alpha - 1}{2(\alpha + 1)}]\\\\
> 1 + \frac{4\eta}{1-2\eta} & \eta \in [\frac{\alpha - 1}{2(\alpha + 1)}, \frac{1}{\alpha}-\frac{1}{2}]\\\\
> 1 + \frac{2\alpha \eta}{\alpha - 1} & \eta \in [\frac{1}{\alpha}-\frac{1}{2}, \frac{1}{2\alpha}] \\\\
> \end{cases}
> $$
> When $\alpha > \frac{1+\sqrt{5}}{2}$, we have the approximation ratio
> $$
> \rho_{\alpha}(\eta)=
> \begin{cases}
> \alpha & \eta \in [0, \frac{(\alpha - 1)^2}{2\alpha}]\\\\
> 1 + \frac{2\alpha\eta}{\alpha - 1} & \eta \in [ \frac{(\alpha - 1)^2}{2\alpha}, \frac{1}{2\alpha}]\\\\
> \end{cases}
> $$
> For both piecewise functions, we have verified that they are continuous and monotonically increasing with respect to the prediction error. Therefore, we argue that **$\alpha$-BIM is smooth with respect to the prediction error.** In the revised version, we will incorporate these results, or at least include a related discussion, to provide insights for interested readers.
>
> For the randomized mechanism, we currently do not have a clear understanding of its smoothness, as deriving an explicit expression for $\rho_{\alpha}(\eta)$ is indeed challenging, and we outline the reasons below. Similar to the smooth analysis of deterministic mechanisms, the randomized one also involves case-by-case examination based on the ranges of $\hat{y}$ and $\eta$. Each case is expressed in the form $\max_j { f_j(\eta, \hat{y}) }$. However, unlike deterministic mechanisms, the presence of $\hat{y}$ in the probability terms introduces significant complexity. Specifically, the numerator of most $f_j(\eta, \hat{y})$ terms includes quadratic expressions like $\hat{y}^2$ and cross terms like $\hat{y} \cdot \eta$, while the denominator involves both $\hat{y}$ and $\eta$. This inherent complexity prevents us from deriving a closed-form expression for $\rho_{\alpha}(\eta)$.
>
> **Q: Randomized mechanism without prediction, the gap between upper bound and lower bound is still big. Any intuition about possible directions to narrow it?**
>
> A: At this stage, we do not have a concrete conjecture regarding how to further narrow the approximation gap for randomized mechanisms. However, we believe that simply extending the support of the mechanism to include more candidate locations with positive probability may not be a promising direction. This intuition is grounded in prior work on facility location problems, where a widely adopted and effective strategy involves selecting three representative points, typically from the left, center, and right of the domain. This approach, exemplified by the well-known LRM mechanism, has been shown to achieve strong approximation guarantees.Our own randomized mechanism without prediction builds upon this three-point idea and successfully breaks the approximation barrier of 2. Moreover, as demonstrated in Theorem 4.3, even optimizing over three candidate locations requires substantial technical effort. Extending the support to more than three locations would likely increase this complexity without yielding clear benefits and may even degrade performance. These considerations reinforce our skepticism about the effectiveness of mechanisms with broader support within the current framework. Therefore, we believe that the most promising direction for future research lies in developing an entirely new framework for randomized mechanism design.
>
> **Q: Missing intuition of $\alpha$-BIM**
>
> A: Thank you for your suggestion. We will incorporate an explanation of the intuition behind the design of $\alpha$-BIM in the revised version of our paper. The key idea is that placing the facility at the predicted location $\hat{y}$ ensures $1$-consistency. However, when $\hat{y}$ approaches the endpoints of the interval (i.e., near $0$ or $1$), this choice becomes increasingly risky in the presence of prediction errors, leading to poor robustness. To address this issue, $\alpha$-BIM is designed to balance consistency and robustness by adopting alternative placement strategies when $\hat{y}$ deviates sufficiently far from the center of the interval. This trade-off helps mitigate the impact of erroneous predictions near the boundaries.
>
> **Q: Minor Typos**
>
> A: We are grateful to the reviewer for pointing out these minor typos. We will correct them in the revised version of our paper.

---

> > ### Comment · Reviewer_gHpJ · 2025-08-01
> >
> > I thank the authors for their response. I believe the smoothness guarantee for the deterministic algorithm and the smoothness discussion for the randomized one would strengthen the paper. I am also satisfied with their responses to the other questions.
> > I will maintain my positive score.

---

### Official Review · Reviewer_jQBn · 2025-06-30

**Clarity:** 3
**Significance:** 3
**Originality:** 3
**Rating:** 4
**Confidence:** 4

**Summary:**

This paper studies the strategic facility location problem in a one-dimensional space. The authors consider a fairness objective, specifically the envy ratio, which is the maximum ratio between the utilities of any two agents.

They examine both deterministic and randomized settings, and they also explore the learning-augmented setting, where some prediction about the optimal outcome is available.
In the deterministic setting, they propose a mechanism and establish a consistency-robustness trade-off, proving its optimality.
In the randomized setting, they improve the best known upper and lower bounds. Moreover, they propose a mechanism with predictions in this setting.

**Questions:**

1- Which techniques or ideas from your paper could potentially extend to more general settings, such as those without the fixed interval constraint or in higher dimensions?

2- Is there any known lower bound for the one-dimensional setting without the fixed interval constraint?

3- Have you considered other prediction settings? For example, can stronger results be obtained if the predictions concern agents' locations?

4- I agree with the authors that establishing a lower bound for randomized mechanisms with predictions is an interesting direction. Could you please elaborate more on the challenges involved?

5- Which of your proposed mechanisms are tolerant to small prediction errors? What guarantees can you provide in terms of smoothness?

**Ethical Concerns:**

["NO or VERY MINOR ethics concerns only"]

**Final Justification:**

This paper addresses an important and practical problem with technically interesting results and overall good writing. While some ideas build on prior work, the contributions are novel and meaningful. My main concern remains the restricted setting, specifically the reliance on a fixed interval, which limits the generality and makes some properties easier to obtain. I appreciate the authors' response and their plan to improve the presentation and include smoothness analysis. However, due to the limitations of the setting, I am not increasing my score.

**Limitations:**

Yes.

**Paper Formatting Concerns:**

No issues.

**Quality:**

3

**Strengths And Weaknesses:**

The facility location problem is a very interesting and practical problem, and I enjoyed reading this work. The quality of writing is good, and the results require significant effort to obtain. Although some of the underlying ideas (such as those in Mechanism 1 or Lemma 3.1) are drawn from previous works, the results do exhibit some novelty.

My main concern is the setting. The results are not only restricted to the line metric, but also to a fixed interval. Without a fixed interval, the constant 1/2 mechanism is not a 2-approximation and can be unbounded on an arbitrary line. The proposed mechanisms rely on this constant 1/2 mechanism, which makes the strategyproofness property relatively straightforward. I recommend that the setting be described more clearly throughout the paper, and that it also be explicitly mentioned in the abstract.

---

> ### Author Rebuttal · Authors · 2025-07-30
>
> Thank you for your insightful comments and suggestions. We respond to your questions and concerns as follows.
>
> **Q: Which techniques or ideas from your paper could potentially extend to more general settings, such as those without the fixed interval constraint or in higher dimensions?**
>
> A: We first clarify that the fixed interval assumption is necessary for our problem setting as it ensures that the utility function remains non-negative and well-defined. Studying higher-dimensional settings is out of scope for our current work, as there is no existing literature that examines envy ratio (without prediction) in high-dimensional facility location problems. Nevertheless, we briefly discuss potential extensions below.
>
> (1). For **deterministic mechanisms**, a natural extension from one-dimensional to higher-dimensional settings involves placing the facility at the centroid of a bounded space, generalizing the interval assumption used in one dimension. However, identifying the centroid in an arbitrary bounded space presents significant challenges due to its geometric complexity. Even in a more specific setting, such as a ball, extending the $\alpha$-BIM mechanism is non-trivial, as the boundaries in higher dimensions extend beyond the two points ($1-1/\alpha$ and $1/\alpha$) used in one dimension, requiring a careful redefinition of the mechanism to account for multidimensional geometry. Overall, we agree that this is an interesting but challenging direction for future work.
>
> (2). For **randomized mechanisms without prediction**, the extension is also challenging. The key difficulty lies in the fact that the LRM approach, relying on three representative points, is not directly applicable in higher dimensions, and more support points would likely be required.
>
> (3). **With prediction**, our proposed $\alpha$-BIM mechanism can, in principle, be extended by considering a bounded subspace. The core idea behind BAM remains applicable: to ensure robustness, the probability of placing the facility at the predicted location should decrease sharply when the prediction is near the boundary of the space. However, it is evident that determining the appropriate probability distribution and analyzing consistency and robustness in this setting pose substantial challenges.
>
> **Q: Is there any known lower bound for the one-dimensional setting without the fixed interval constraint?**
>
> A: We clarify that the fixed interval constraint is essential for properly defining each agent’s utility. Moreover, for the envy ratio objective, a utility-based formulation is necessary. If a distance-based definition were used instead, the envy ratio would become unbounded whenever the facility is located at an agent’s position.
>
> **Q: Have you considered other prediction settings? For example, can stronger results be obtained if the predictions concern agents' locations?**
>
> A:  For deterministic mechanisms with prediction, we establish tight bounds under the assumption of optimal location prediction. Since the optimal location can be computed directly from agents’ reported predictions, this implies that access to an optimal predictor is sufficient for the design of effective deterministic mechanisms. For randomized mechanisms, Lemma 3.1 shows that the analysis of the envy ratio approximation can be reduced to two-agent instances. This insight suggests that incorporating predictions about the positions of the leftmost and rightmost agents could potentially yield stronger approximation guarantees by more fine-grained analysis. However, investigating such extensions is beyond the scope of the current paper and represents an interesting direction for future research.
>
> **Q: What is the challenge of establishing the lower bound for randomized mechanisms with prediction?**
>
> A: Thank you for your question. We would like to highlight the difficulty of establishing lower bounds for randomized mechanisms with predictions under our objective. For classical objectives such as social cost and maximum cost, the paper “Randomized Strategic Facility Location with Predictions” successfully derives lower bounds by elegantly characterizing all mechanisms within a class called “ONLYM mechanisms,” and then proving bounds specific to that class. Motivated by their approach, we attempted to develop a similar characterization tailored to our setting. However, we encountered significant obstacles in identifying a corresponding class of mechanisms suitable for our objective. Unlike social cost or maximum cost, their objective function is linear, while ours is not. In particular, our objective can be seen as a multiplicative ratio, with the numerator involving a maximum utility function and the denominator involving a minimum utility function, which introduces significant analytical complexity. This structure makes precise characterization challenging, as it requires identifying the agents responsible for the maximum and minimum utilities and accounting for how these agents may change when the facility’s location is adjusted.
>
> **Q: Which of your proposed mechanisms are tolerant to small prediction errors? What guarantees can you provide in terms of smoothness?**
>
> A: Thank you for raising the issue regarding smoothness analysis. We initially derived results for the approximation ratio with respect to prediction error. However, we chose not to include them in the main text, as their analysis is sufficiently complex and the results are less intuitive than the (consistency, robustness) framework. In the revised version, we will incorporate these results, or at least include a related discussion, to provide insights for interested readers.
>
> For the deterministic mechanism with prediction, $\alpha$-BIM,  let $\eta$ be the prediction error. Notice that the mechanism $\alpha$-BIM is parameterized by $\alpha$, We have the following approximation ratio result.
> When $\alpha \leq \frac{1+\sqrt{5}}{2}$, we have the approximation ratio
> $$
> \rho_{\alpha}(\eta)=
> \begin{cases}
> \alpha & \eta \in [0, \frac{\alpha - 1}{2(\alpha + 1)}]\\\\
> 1 + \frac{4\eta}{1-2\eta} & \eta \in [\frac{\alpha - 1}{2(\alpha + 1)}, \frac{1}{\alpha}-\frac{1}{2}]\\\\
> 1 + \frac{2\alpha \eta}{\alpha - 1} & \eta \in [\frac{1}{\alpha}-\frac{1}{2}, \frac{1}{2\alpha}] \\\\
> \end{cases}
> $$
> When $\alpha > \frac{1+\sqrt{5}}{2}$, we have the approximation ratio
> $$
> \rho_{\alpha}(\eta)=
> \begin{cases}
> \alpha & \eta \in [0, \frac{(\alpha - 1)^2}{2\alpha}]\\\\
> 1 + \frac{2\alpha\eta}{\alpha - 1} & \eta \in [ \frac{(\alpha - 1)^2}{2\alpha}, \frac{1}{2\alpha}]\\\\
> \end{cases}
> $$
> Intuitively, Assume that we know the prediction model has an error smaller than $\frac{\sqrt{5}}{2} - 1\approx 0.118$, then we have the approximation ratio of $\frac{\sqrt{5}+1}{2}\approx 1.62$. This implies that **when we have a good prediction model with bounded small error. We can significantly improve the approximation ratio from 2 to 1.62 by our proposed $\alpha$-BIM**.
>
> For the randomized mechanism, unfortunately, providing an explicit expression for $\rho_{\alpha}(\eta)$ is indeed difficult, and we outline the reasons below. Similar to the smooth analysis of deterministic mechanisms, the randomized one also involves case-by-case examination based on the ranges of $\hat{y}$ and $\eta$. Each case is expressed in the form $\max_j { f_j(\eta, \hat{y}) }$. However, unlike deterministic mechanisms, the presence of $\hat{y}$ in the probability terms introduces significant complexity. Specifically, the numerator of most $f_j(\eta, \hat{y})$ terms includes quadratic expressions like $\hat{y}^2$ and cross terms like $\hat{y} \cdot \eta$, while the denominator involves both $\hat{y}$ and $\eta$. This inherent complexity prevents us from deriving a closed-form expression for $\rho_{\alpha}(\eta)$.

---

> > ### Comment · Reviewer_jQBn · 2025-08-04
> >
> > Thank you for your response and for elaborating on the challenges. I agree that including a smoothness analysis would be a valuable addition to the paper. While I appreciate the idea, I was unable to verify the correctness, particularly because the definition of the error measure is missing. I have no further questions. Due to the limitations of the setting, I will not raise my score.

---

### Official Review · Reviewer_P6NH · 2025-07-02

**Clarity:** 3
**Significance:** 3
**Originality:** 3
**Rating:** 5
**Confidence:** 3

**Summary:**

This paper studies the facility location problem for the envy ratio objective, with and without machine learning predictions. The envy ratio is a fairness metric defined as the maximum ratio of utilities between any two agents. The goal is to find anonymous and strategy proof mechanisms that have small approximation ratios. For the setting without machine learning predictions, this paper proposes a randomized algorithm $(\alpha, p)$-LRM, which improves the approximation ratio from 2 (by Ding et al. [2020]) to 1.8944. The authors also give a lower bound example which improves the previous lowder bound of 1.0314 to 1.12579. For the setting with machine learning predictions, the authors propose a learning augmented deterministic algorithm with $\alpha$-consistency and $\frac{\alpha}{\alpha-1}$ -robustness, with $\alpha \in [1, 2]$, and show the optimality of this algorithm. Finally, they propose a learning augmented randomized mechanism BAM that outperforms the $\alpha$-BIM in both consistency and robustness.

**Questions:**

* In Theorem 4.5, does the robustness proof work for any $c \ in [0, \frac{1}{2}]$? Why is the maximum robustness for $c \ in [0, \frac{1}{4})$ = $\frac{7}{4}$ instead of $\frac{9}{4}$?
* Also for Theorem 4.5, why do you state the maximum consistency and robustness for $c\in[0,\frac{1}{4})$ instead of formulas with $c$ as a parameter?
* There is still a large gap between the upper and lower bounds for the randomized mechanisms without prediction. Do you have any conjecture about what type of algorithms could improve the approximation ratio? Does it help to return more than three locations?

**Ethical Concerns:**

["NO or VERY MINOR ethics concerns only"]

**Final Justification:**

This paper appropriately builds upon prior research in facility location mechanisms for the envy ratio objective and learning-augmented algorithms. The authors have addressed all my questions. The main remaining concern is the relatively large gap between the upper and lower bounds, but I think the problem and techniques proposed in this paper will have positive impact on the theoretical research on facility location problems.

**Limitations:**

The authors discussed the limitations and future directions.

**Quality:**

3

**Strengths And Weaknesses:**

* Strengths
    * The paper is well-written and organized, with clearly presented and proven theorems and statements.
    * This work appropriately builds upon prior research in facility location mechanisms for the envy ratio objective and learning-augmented algorithms. All related works are properly cited.
    * The paper enhances the upper and lower bounds of the randomized mechanism without prediction. Additionally, it introduces novel mechanisms that offer consistency-robustness tradeoffs in the setting with machine learning predictions. The analysis and lower bound constructions are non-trivial.
* Weaknesses
    * A direct future direction is to reduce the gap between the upper and lower bounds for randomized mechanisms without prediction.

* Minor comments
    * Page 19, line 757. “$u(y, x_1) = 1 - d(y, x_i)$” => $x_i$ should be $x_1$
    * Page 20, line 789 and 791. “$\rho(\mathbf{x})$ is monotonically non-increasing / increasing” Do you mean the upper bound of $\rho(\mathbf{x})$?

---

> ### Author Rebuttal · Authors · 2025-07-30
>
> Thank you for your insightful comments and suggestions. We respond to your questions and concerns as follows.
>
> **Q: The typo of the robustness in Theorem 4.5. Why state the maximum consistency and robustness for $c\in [0, \frac{1}{4})$ instead of formulas with $c$ as a parameter?**
>
> A: Thank you for pointing out the typo in Theorem 4.5. We acknowledge that it is a typo and the correct value should be $\frac{9}{4}$ rather than $\frac{7}{4}$. This aligns with our Case 1 analysis, which is continuous, as thoroughly detailed in the proof and illustrated in Figure 2, representing the worst-case scenario. We will make the correction in the revised version of the paper. Secondly, we clarify that in BAM, $c$ is not an input parameter. When $c\in [0,\frac{1}{4})$, BAM achieves $\frac{7}{4}$- consistency and $\frac{9}{4}$-robustness as these values correspond to the worst-case ratios derived from our consistency and robustness analysis. For $c\in [\frac{1}{4}, \frac{1}{2})$, we express the performance of BAM as $(-4c^2+2)$-consistency and $(c+2)$-robustness. This is because in this range, the worst cases for consistency and robustness are no longer directly comparable, and the trade-off between the two defines a Pareto frontier. As $c$ increases from $\frac{1}{4}$ to $\frac{1}{2}$,the performance of BAM smoothly transitions from $1$-consistency and $\frac{5}{2}$-robustness to $\frac{7}{4}$-consistency and $\frac{9}{4}$-robustness.
>
> **Q: Any idea or conjecture of narrowing the gap for randomized mechanisms without prediction? For example, considering more than three candidate locations?**
>
> A: We consider that developing a novel framework for randomized mechanisms, distinct from the one presented in this work, offers greater potential and appeal than increasing the number of candidate locations within the existing framework, though we do not have a concrete conjecture on what the new framework might entail. This intuition is grounded in prior work on facility location problems, where a widely adopted and effective strategy involves selecting three representative points, typically from the left, center, and right of the domain. This approach, exemplified by the well-known LRM mechanism, has been shown to achieve strong approximation guarantees.Our own randomized mechanism without prediction builds upon this three-point idea and successfully breaks the approximation barrier of 2. Moreover, as demonstrated in Theorem 4.3, even optimizing over three candidate locations requires substantial technical effort. Extending the support to more than three locations would likely increase this complexity without yielding clear benefits and may even degrade performance. These considerations reinforce our skepticism about the effectiveness of mechanisms with broader support within the current framework. Therefore, we believe that the most promising direction for future research lies in developing an entirely new framework for randomized mechanism design.
>
> **Q: Minor comments.**
>
> A: Thank you for pointing out the typos. We will make the corresponding edits in the revised version of our paper.

---

> > ### Comment · Reviewer_P6NH · 2025-08-01
> >
> > Thank you for your response! My questions are addressed.

---

### Decision · Program_Chairs · 2025-09-17

**Decision:**

Accept (poster)

**Comment:**

The paper studies strategyproof mechanisms for facility location games, focusing on the envy ratio objective, in the learning-augmented framework.  The authors present improved upper and lower bounds in a number of settings.  The reviewers find the results meaningful and nontrivial.  The main concern is the fairly restricted setting, which ultimately is not considered a disqualifying weakness.  We encourage the authors to further improve the paper based on the constructive comments.